# VIMA: General Robot Manipulation with Multimodal Prompts

## Abstract

Prompt-based learning has emerged as a successful paradigm in natural language processing, where a single general-purpose language model can be instructed to perform any task specified by input prompts. Yet task specification in robotics comes in various forms, such as imitating one-shot demonstrations, following language instructions, and reaching visual goals. They are often considered different tasks and tackled by specialized models. This work shows that we can express a wide spectrum of robot manipulation tasks with *multimodal prompts*, interleaving textual and visual tokens. We design a transformer-based robot agent, VIMA, that processes these prompts and outputs motor actions autoregressively. To train and evaluate VIMA, we develop a new simulation benchmark with thousands of procedurally-generated tabletop tasks with multimodal prompts, 600K+ expert trajectories for imitation learning, and four levels of evaluation protocol for systematic generalization. VIMA achieves strong scalability in both model capacity and data size. It outperforms prior SOTA methods in the hardest zero-shot generalization setting by up to $2.9\times$ task success rate given the same training data. With $10\times$ less training data, VIMA still performs $2.7\times$ better than the top competing approach. Video demos are available at https://iclr3081.github.io/.

## 1 Introduction

Transformers have given rise to remarkable multi-task consolidation across many AI domains. For example, users can describe a task using natural language prompt to GPT-3 (Brown et al., 2020), allowing the same model to perform question answering, machine translation, text summarization, etc. Prompt-based learning provides an accessible and flexible interface to communicate a natural language understanding task to a general-purpose model.

We envision that a generalist robot agent should have a similarly intuitive and expressive interface for task specification. What does such an interface for robot learning look like? As a motivating example, consider a personal robot tasked with household activities. We can ask the robot to bring us a cup of water by a simple natural language instruction. If we require more specificity, we can instead instruct the robot to "bring me <image of the cup>". For tasks requiring new skills, the robot should be able to adapt preferably from a few video demonstrations (Duan et al., 2017). Tasks that need interaction with unfamiliar objects can be easily explained via a few image examples for *novel concept grounding* (Hermann et al., 2017). Finally, to ensure safe deployment, we can further specify visual constraints like "do not enter <image> room".

To enable a single agent with all these capabilities, we make three key contributions in this work: 1) a novel **multimodal prompting formulation** that converts a wide spectrum of robot manipulation tasks into one sequence modeling problem; 2) a new **robot agent model** capable of multi-task and zero-shot generalization; and 3) a **large-scale benchmark** with diverse tasks to systematically evaluate the scalability and generalization of our agents.

We start with the observation that many robot manipulation tasks can be formulated by **multimodal prompts that interleave language and images or video frames** (Fig. 1). For example, Rearrangement (Batra et al., 2020), a type of *Visual Goal*, can be formulated as "Please rearrange objects to match this {scene_image}"; *Novel Concept Grounding* looks like "This is a dax {new_object}₁ and this is a blicket {new_object}₂. Put two metal dax on the marble blicket."; *Few-shot Imitation* can embed video snippet in the prompt "Follow this motion trajectory for the wooden cube:

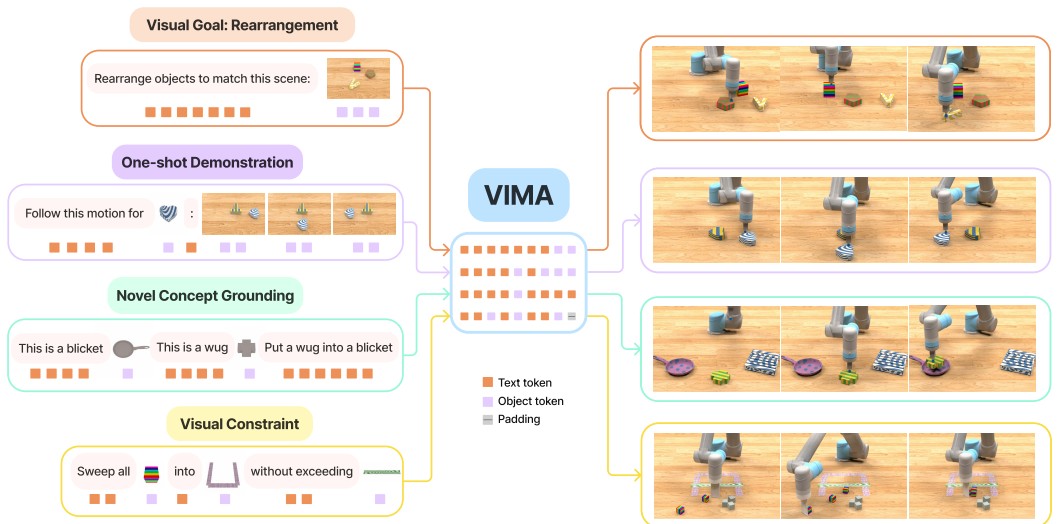

Figure 1: **Multimodal prompts for task specification.** We observe that many robot manipulation tasks can be expressed as *multimodal prompts* that interleave language and image/video frames. We propose VIMA, an embodied agent model capable of processing mulitimodal prompts (left) and controlling a robot arm to solve the task (right).

{frame$_1$}, {frame$_2$}, {frame$_3$}, {frame$_4$}"; and expressing *Visual Constraint* is as simple as adding the clause "without touching {safety_boundary}".

Multimodal prompts not only have more expressive power than individual modalities, but also enable a **uniform sequence IO interface** for training generalist robot agents. Previously, different robot manipulation tasks require distinct policy architectures, objective functions, data pipelines, and training procedures (Aceituno et al., 2021; Stengel-Eskin et al., 2022; Lynch & Sermanet, 2021), leading to siloed robot systems that cannot be easily combined for a rich set of use cases. Instead, our multimodal prompt interface allows us to harness the latest advances in large transformer models (Lin et al., 2021; Tay et al., 2020; Khan et al., 2021) for developing scalable multi-task robot learners.

To this end, we design a novel **Vi**suo**M**otor **A**ttention model (VIMA). The architecture follows the encoder-decoder transformer design proven to be effective and scalable in NLP (Raffel et al., 2020). VIMA encodes an input sequence of interleaving textual and visual prompt tokens with a pre-trained language model (Tsimpoukelli et al., 2021), and decodes robot control actions autoregressively for each environment interaction step. The transformer decoder is conditioned on the prompt via cross-attention layers that alternate with the usual causal self-attention. Instead of operating on raw pixels, VIMA adopts an object-centric approach. We parse all images in the prompt or observation into objects by off-the-shelf detectors (He et al., 2017), and flatten them into sequences of object tokens. All these design choices combined deliver a conceptually simple architecture with strong model and data scaling properties.

To systematically evaluate our proposed algorithm, we introduce a new benchmark (VIMA-BENCH) built on the Ravens simulator (Zeng et al., 2020; Shridhar et al., 2021). We provide 17 representative meta-tasks with multimodal prompt templates, which can be procedurally instantiated into thousands of individual tasks by various combinations of textures and tabletop objects. VIMA-BENCH establishes a 4-level protocol to evaluate progressively stronger generalization capabilities, from randomized object placement to novel tasks altogether (Fig. 2). To demonstrate the scalability of VIMA, we train a spectrum of 7 models ranging from 2M to 200M parameters. Our approach outperforms strong prior SOTA methods such as Gato (Reed et al., 2022), Decision Transformer (Chen et al., 2021), and Flamingo (Alayrac et al., 2022) across all 4 levels of zero-shot generalization and all model capacities, sometimes by a large margin (up to $2.9\times$ task success rate given the same amount of training data, and $2.7\times$ better even with $10\times$ less data). We plan to open-source the simulation environment, training dataset, algorithm code, and pre-trained model checkpoints to ensure reproducibility and facilitate future works from the community. We attach supplementary materials as Appendix to this PDF, and present video demos and anonymized code at https://iclr3081.github.io/.

## 2 RELATED WORK

**Multi-task Learning by Sequence Modeling.** Transformers have enabled task unification across many AI domains (Raffel et al., 2020; Brown et al., 2020; Chen et al., 2022a;b; Lu et al., 2022; Wang et al., 2022c; Alayrac et al., 2022). For example, in **NLP**, T5 (Raffel et al., 2020) unifies all language problems into the same text-to-text format. GPT-3 (Brown et al., 2020), PaLM (Chowdhery et al., 2022), and Megatron (Shoeybi et al., 2019) demonstrate emergent behaviours of intuitive task specifications by zero-shot prompting. In **computer vision**, Florence (Yuan et al., 2021), BiT (Kolesnikov et al., 2020), and MuST (Ghiasi et al., 2021) pre-train a shared backbone model at scale for general visual representations and transfer it to downstream tasks. Pix2Seq (Chen et al., 2022b) casts many vision problems into a unified sequence format. In **multimodal learning**, Flamingo (Alayrac et al., 2022) and Frozen (Tsimpoukelli et al., 2021) design a universal API that ingests an interleaving sequence of images and text and generates free-form text. Gato (Reed et al., 2022) is a massively multi-task model across NLP, vision, and embodied agents. Our work is most similar in spirit to Gato, but we focus primarily on enabling an intuitive, multimodal prompting interface for a generalist robot agent.

**Foundation Models for Embodied Agents.** Foundation models (Bommasani et al., 2021; Brown et al., 2020; Raffel et al., 2020; Ramesh et al., 2022; Wei et al., 2022) have demonstrated strong emergent properties like zero-shot prompting and complex reasoning. There are many ongoing efforts to replicate this success for embodied agents, focusing on 3 aspects: 1) **Transformer agent architecture**: Decision Transformer (Chen et al., 2021; Janner et al., 2021; Zheng et al., 2022; Xu et al., 2022) and Gato (Reed et al., 2022) leverage the powerful self-attention models for sequential decision making. CLIPort (Shridhar et al., 2021) and Perceiver-Actor (Shridhar et al., 2022) apply large transformers to robot manipulation tasks. Methods such as Dasari & Gupta (2020) and MOSAIC (Zhao et al., 2022) also leverage transformers to achieve superior performance in one-shot video imitation tasks. 2) **Pre-training for better representations**: Masked ViT (Gupta et al., 2022b), R3M (Nair et al., 2022), and Parisi et al. (2022) pre-train general visual representations for robotic perception. Li et al. (2022); Reid et al. (2022) finetune from LLM checkpoints to accelerate policy learning. MineDojo (Fan et al., 2022) and Ego4D (Grauman et al., 2021) provide large-scale multimodal databases to facilitate scalable policy training. 3) **Large language models for robot learning**: SayCan (Ahn et al., 2022) leverages the 500B PaLM (Chowdhery et al., 2022) for zero-shot concept grounding. Socratic Models (Zeng et al., 2022) composes multiple vision and language foundation models (VLMs) for multimodal reasoning in videos. Huang et al. (2022a), Inner Monologue (Huang et al., 2022b) and LM-Nav (Shah et al., 2022) successfully apply LLMs to long-horizon robot planning. VIMA differs from these works in our novel multimodal prompting formulation, which existing LLMs and VLMs do not easily support.

**Robot Manipulation and Benchmarks.** There are a wide range of robot manipulation tasks that require different skills and task specification formats, such as instruction following (Stepputtis et al., 2020; Shridhar et al., 2021; Lynch & Sermanet, 2021), one-shot imitation (Finn et al., 2017; Dasari & Gupta, 2020; Duan et al., 2017), rearrangement (Batra et al., 2020; Weihs et al., 2021; Szot et al., 2021), constraint satisfaction (Brunke et al., 2021a; Srinivasan et al., 2020; Thananjeyan et al., 2021), and reasoning (Shridhar et al., 2020; Gupta et al., 2019; Ahmed et al., 2021; Toyer et al., 2020; Lim et al., 2021). Multiple physics simulation benchmarks are introduced to study the above tasks. For example, iGibson (Shen et al., 2020; Li et al., 2021; Srivastava et al., 2021) simulates interactive household scenarios. Ravens (Zeng et al., 2020) and Robosuite (Zhu et al., 2020; Fan et al., 2021) design various tabletop manipulation tasks with realistic robot arms. MOSAIC (Zhao et al., 2022) features a challenging benchmark built on top of Zhu et al. (2020) for one-shot imitation learning. Our VIMA-BENCH is the first robot learning benchmark to support multimodal-prompted tasks. We also standardize the evaluation protocol to systematically measure an agent's generalization capabilities.

A more extended literature review can be found in Appendix, Sec. F.

## 3 MULTIMODAL PROMPTS FOR TASK SPECIFICATION

A central and open problem in robot learning is task specification (Agrawal, 2022). In prior literature (Stepputtis et al., 2020; Dasari & Gupta, 2020; Brunke et al., 2021b), different tasks often require diverse and incompatible interfaces, resulting in siloed robot systems that do not generalize well across tasks. Our key insight is that various task specification paradigms (such as goal conditioning,

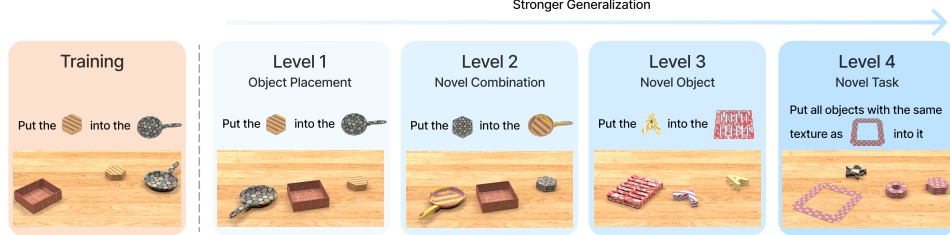

Figure 2: **Evaluation Protocol in VIMA-BENCH**. We design 4 levels of evaluation settings to measure the zero-shot generalization capability of an agent systematically. Each level deviates more from the training distribution, and thus is strictly more challenging than the previous level.

video demonstration, natural language instruction) can all be instantiated as multimodal prompts (Fig. 1). Concretely, a multimodal prompt $\mathcal{P}$ of length $l$ is defined as an ordered sequence of arbitrarily interleaved texts and images $\mathcal{P} := [x_0, x_1, \ldots, x_l]$, where each element $x_i \in \{\text{text}, \text{image}\}$.

**Task Suite.** The flexibility afforded by multimodal prompts allows us to specify and build models for a huge variety of task specification formats. Here we consider the following six task categories. 1. **Simple object manipulation**: simple tasks like "put <object> into <container>", where each image in the prompt corresponds to a single object; 2. **Visual goal reaching**: manipulating objects to reach a goal configuration, *e.g.*, *Rearrangement* (Batra et al., 2020); 3. **Novel concept grounding**: the prompt contains unfamiliar words like "dax" and "blicket", which are explained by in-prompt images and then immediately used in an instruction. This tests the agent's ability to rapidly internalize new concepts; 4. **One-shot video imitation**: watching a video demonstration and learning to reproduce the same motion trajectory for a particular object; 5. **Visual constraint satisfaction**: the robot must manipulate the objects carefully and avoid violating the (safety) constraints; 6. **Visual reasoning**: tasks that require reasoning skills, such as appearance matching "move all objects with same textures as <object> into a container", and visual memory, "put <object> in container and then restore to their original position".

Note that these six categories are not mutually exclusive. For example, a task may introduce a previously unseen verb (*Novel Concept*) by showing a video demonstration, or combine goal reaching with visual reasoning. More details about the task suite are discussed in Appendix, Sec. B.

## 4 VIMA-BENCH: BENCHMARK FOR MULTIMODAL ROBOT LEARNING

**Simulation Environment.** Existing benchmarks are generally geared towards a particular task specification. To our knowledge, there is no benchmark that provides a rich suite of multimodal tasks and a comprehensive testbed for targeted probing of agent capabilities. To this end, we introduce a new benchmark suite for multimodal robot learning that we call VIMA-BENCH. We built our benchmark by extending the Ravens robot simulator (Zeng et al., 2020). VIMA-BENCH supports extensible collections of objects and textures to compose multimodal prompts and procedurally generate a large number of tasks. Specifically, we provide 17 meta-tasks with multimodal prompt templates, which can be instantiated into 1000s of individual tasks. Each meta-task belongs to one or more of 6 task categories mentioned above. VIMA-BENCH can generate large quantities of imitation learning data via scripted oracle agents. More details are elaborated in Appendix, Sec. A.

**Observation and Actions.** The observation space of our simulator includes RGB images rendered from both frontal view and top-down view. Groundtruth object segmentations and bounding boxes are also provided for training object-centric models (Sec. 5). We inherit the high-level action space from Zeng et al. (2020), which consists of primitive motor skills like "pick and place" and "wipe". These are parameterized by poses of the end effector. Our simulator also features scripted oracle programs that can generate expert demonstrations by using privileged simulator state information, such as the precise location of all objects, and the groundtruth interpretation of the multimodal instruction.

**Training Dataset.** We leverage the pre-programmed oracles to generate a large offline dataset of expert trajectories for imitation learning. Our dataset includes 50K trajectories per meta-task, and 650K successful trajectories in total. We hold out a subset of object models and textures for evaluation, and designate 4 out of 17 meta-tasks as a testbed for zero-shot generalization.

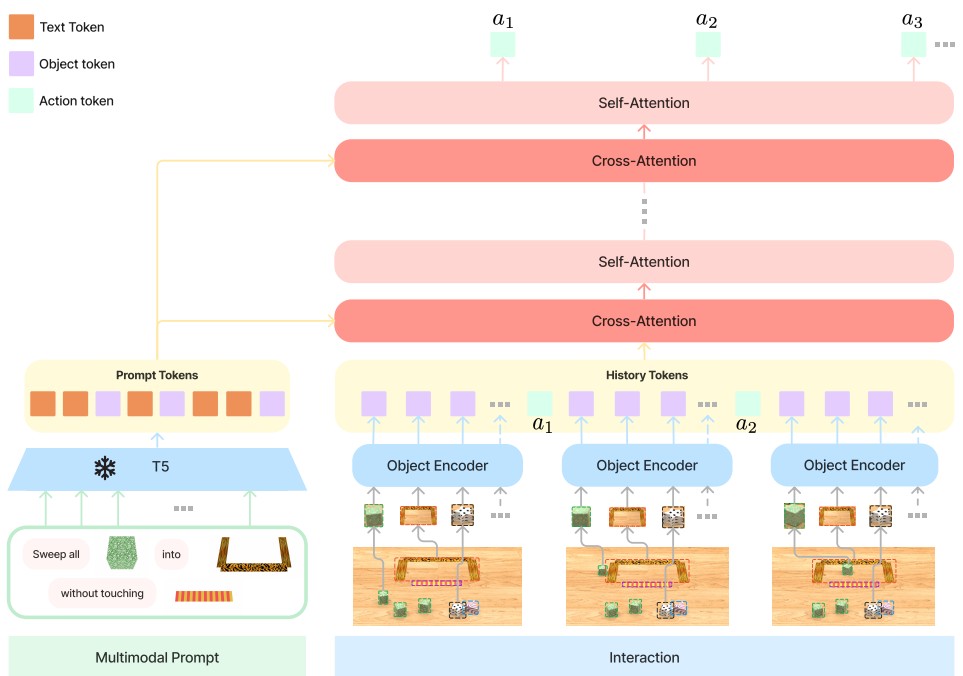

Figure 3: **VIMA.** We encode the multimodal prompts with a pre-trained T5 model, and condition the robot controller on the prompt through cross-attention layers. The controller is a causal transformer decoder consisting of alternating self and cross attention layers that predicts motor commands conditioned on prompts and interaction history.

**Evaluating Zero-Shot Generalization.** Each task in VIMA-BENCH has a binary success criterion and does not provide partial reward signals. During test time, we execute the agent policies in the physics simulator for multiple episodes to compute a success rate in percentage. The average success rate over all evaluated meta-tasks will be the final reported metric.

We design a 4-level evaluation protocol (Fig. 2) to systematically probe the generalization capabilities of learned agents. Each level deviates more from the training distribution, and is thus strictly harder than the previous one — Level 1) **placement generalization**: all prompts are seen verbatim during training, but only the placement of objects on the tabletop is randomized at testing; Level 2) **combinatorial generalization**: all materials (adjectives) and 3D objects (nouns) are seen during training, but new combinations of them appear in testing; Level 3) **novel object generalization**: test prompts and the simulated workspace include novel adjectives and objects; Level 4) **novel task generalization**: new meta-tasks with novel prompt templates at test time.

## 5 VIMA: VISUOMOTOR ATTENTION MODEL

Our goal is to build a robot agent capable of performing any task specified by multimodal prompts. To learn an effective multi-task robot policy, we propose VIMA, a minimalistic multi-task encoder-decoder architecture with object-centric design (Fig. 3). Concretely, we learn a robot policy $\pi(a_t | \mathcal{P}, \mathcal{H})$, where $\mathcal{H} := [o_1, a_1, o_2, a_2, \ldots, o_t]$ denotes the past interaction history, and $o_t \in \mathcal{O}, a_t \in \mathcal{A}$ are observations and actions at each interaction steps. We encode multimodal prompts via a *frozen* pre-trained langauge model and decode robot motor commands conditioned on the encoded prompts via cross-attention layers. Unlike prior works (Florence et al., 2019; Sieb et al., 2019), VIMA adopts an object-centric token representation that computes features from bounding box coordinates and cropped RGB patches.

**Tokenization.** There are 3 formats of raw input in the prompt — text, image of a single object, and image of a full tabletop scene (*e.g.*, for *Rearrangement* or imitation from video frames). For **text inputs**, we use pre-trained T5 tokenizer and word embedding to obtain word tokens. For **images of full scenes**, we first extract individual objects using off-the-shelf Mask R-CNN (He et al., 2017). Each object is represented as a bounding box and a cropped image. We then compute object tokens by encoding them with a bounding box encoder and a ViT, respectively. Since Mask-RCNN is imperfect,

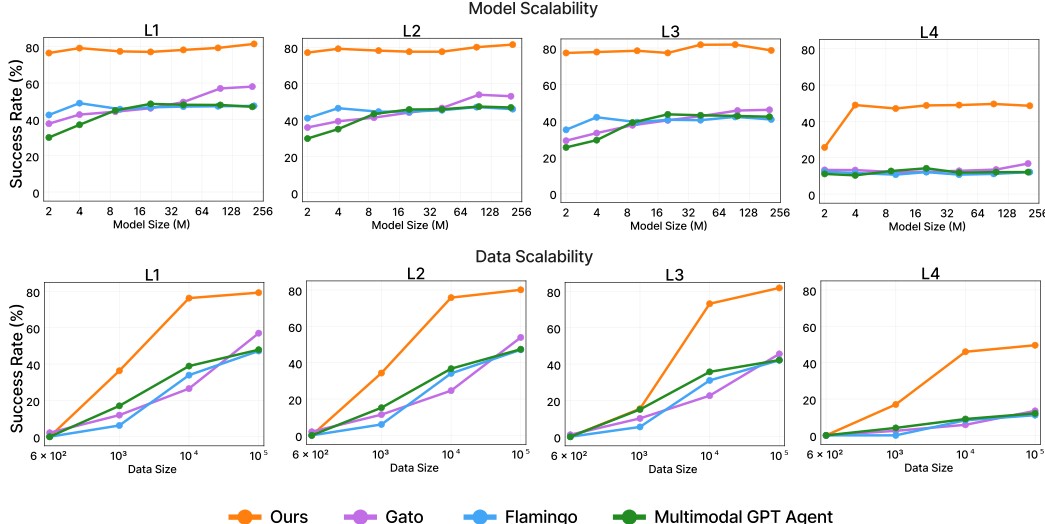

Figure 4: **Scaling model and data**. *Top:* We compare performance of different methods with model sizes ranging from 2M to 200M parameters. Across all model sizes and generalization levels VIMA outperforms prior works. *Bottom:* For a fixed model size of 92M parameters we compare the effect of imitation learning dataset size of $0.1\%$, $1\%$, $10\%$, and full imitation data. VIMA is extremely sample efficient and can achieve performance comparable to other methods with $10\times$ less data.

the bounding boxes can be noisy and the cropped image may have irrelevant pixels. For **images of single objects**, we obtain tokens in the same way except with a dummy bounding box. Prompt tokenization produces a sequence of interleaved textual and visual tokens. We then follow the practice in Tsimpoukelli et al. (2021) and encode the prompt via a pre-trained T5 encoder (Raffel et al., 2020). Since T5 has been pre-trained on large text corpora, VIMA inherits the semantic understanding capability and robustness properties. To accommodate tokens from new modalities, we insert MLPs between the non-textual tokens and T5. To prevent catastrophic forgetting, VIMA finetunes the last two layers of the language encoder with layer-wise learning rate decay (He et al., 2021) but freezes all other layers. Our positional embedding is learnable and absolute.

**Robot Controller.** A challenging aspect of designing multi-task policy is to select a suitable conditioning mechanism. In our schema (Fig. 3), the robot controller (decoder) is conditioned on the prompt sequence $\mathcal{P}$ by a series of cross-attention layers between $\mathcal{P}$ and the trajectory history sequence $\mathcal{H}$. We compute key $K_{\mathcal{P}}$ and value $V_{\mathcal{P}}$ sequences from the prompt and query $Q_{\mathcal{H}}$ from the trajectory history, following the encoder-decoder convention in T5 (Raffel et al., 2020). Each cross-attention layer then generates an output sequence $\mathcal{H}' = \text{softmax}\left(\frac{Q_{\mathcal{H}} K_{\mathcal{P}}^{\mathsf{T}}}{\sqrt{d}}\right) V_{\mathcal{P}}$, where $d$ is the embedding dimension. Residual connections (He et al., 2015) are added to connect higher layers with the input rollout trajectory sequence. The cross-attention design enjoys three advantages: 1) strengthened connection to prompt; 2) intact and deep flow of the original prompt tokens; and 3) better computational efficiency, as demonstrated in VideoGPT (Yan et al., 2021) as well. VIMA decoder consists of $L$ alternating cross-attention and self-attention layers. Finally, we follow common practice (Baker et al., 2022) to map predicted action tokens to discretized coordinates of the robot arm. See Appendix, Sec. C.2 for more details.

**Training.** We follow behavioral cloning to train our models by minimizing the negative log-likelihood of predicted actions. Concretely, for a trajectory with $T$ steps, we minimize $\min_\theta \sum_{t=1}^{T} -\log \pi_\theta(a_t|\mathcal{P}, \mathcal{H})$. The entire training is conducted on an offline dataset with no simulator access. To make VIMA robust to detection inaccuracies and failures, we apply *object augmentation* by randomly injecting *false-positive* detection outputs. After training, we select model checkpoints for evaluation based on the aggregated accuracy on a held-out validation set. The evaluation involves interacting with the physics simulator. We follow the best practices to train Transformer models using the AdamW optimizer (Loshchilov & Hutter, 2019), learning rate warm-up, cosine annealing (Loshchilov & Hutter, 2016), etc. See Appendix Sec. D for comprehensive training hyperparameters.

## 6 EXPERIMENTS

In this section, we aim to answer three main questions: (1) How does VIMA compare with prior SOTA transformer-based agents on a diverse collection of multimodal-prompted tasks? (2) What are the **scaling properties** of our approach in model capacity and data size? (3) How do different visual tokenizers, prompt conditioning, and prompt encoding affect decision making?

### 6.1 BASELINES

**Gato** (Reed et al., 2022) introduces a decoder-only model that solves tasks from multiple domains where tasks are specified by prompting the model with the observation and action subsequence. For fair comparison, we provide the same conditioning as VIMA, *i.e.*, our multimodal embedded prompt. Input images are divided into patches and encoded by a ViT (Dosovitskiy et al., 2020) model to produce observation tokens.

**Flamingo** (Alayrac et al., 2022) is a vision-language model that learns to generate textual completion in response to multimodal prompts. It embeds a variable number of prompt images into a fixed number of tokens via a Perceiver Resampler (Jaegle et al., 2021b), and conditions the language decoder on the encoded prompt by cross-attention. Flamingo does not work with embodied agents out of the box. We adapt it to support decision-masking by replacing the output layer with robot action heads.

**Multimodal GPT agent** is a GPT-based behavior cloning agent conditioned on tokenized multimodal prompts. It autoregressively decodes next actions given instructions and interaction histories. Similar to prior works of casting RL problems as sequence modeling (Chen et al., 2021; Janner et al., 2021), it encodes an image into a single *state* token by a ViT encoder, and prepends the rollout trajectory with prompt tokens. This baseline does not involve cross-attention.

A more detailed comparison between these methods can be found in Appendix, Sec. C.1.

### 6.2 EVALUATION RESULTS

We compare VIMA against other SOTA methods on the four levels of generalization provided in our benchmark for different model and training dataset sizes.

**Model scaling.** We train all methods for a spectrum of model capacities from 2M to 200M parameters, evenly spaced on the log scale. The encoder size is kept constant (pre-trained T5-Base) for all methods and excluded from the parameter count. Across *all* levels of zero-shot generalization, we find that VIMA strongly outperforms prior work. Although models like Gato and Flamingo show improved performance with bigger model sizes, VIMA consistently achieves superior performance over *all* model sizes. We note that this can only be achieved with *both* cross-attention and object token sequence representation without any downsampling — altering any component will degrade the performance significantly, especially in the low model capacity regime (ablations in Sec. 6.3).

**Data scaling.** Next we investigate how different methods scale with varying dataset sizes. We compare model performance at $0.1\%$, $1\%$, $10\%$ and full imitation learning dataset provided in VIMA-BENCH (Fig. 4). VIMA is extremely sample efficient and with just $1\%$ of the data can achieve performance similar to baseline methods trained with $10\times$ more data for L1 and L2 levels of generalization. In fact, for L4 we find that with just $1\%$ of training data, VIMA already outperforms prior work trained with *entire* dataset. Finally, across all levels with just $10\%$ of the data, VIMA can outperform prior work trained with the full dataset by a significant margin. We hypothesize that the data efficiency can be attributed to VIMA's object-centric representation, which is less prone to overfitting than learning directly from pixels in the low-data regime. This is consistent with findings from Sax et al. (2018), which demonstrates that embodied agents conditioned on mid-level visual representations tend to be significantly more sample-efficient than end-to-end control from raw pixels.

**Progressive Generalization.** Finally, we compare the relative performance degradation as we test the models on progressively challenging zero-shot evaluation levels without further finetuning (Fig. 5). Our method exhibits a minimal performance regression, especially between $L1 \to L2$ and $L1 \to L3$. In contrast, other methods can degrade as much as $20\%$, particularly in more difficult generalization scenarios. Although all methods degrade significantly when evaluated on $L4$ (*Novel Tasks*), the drop

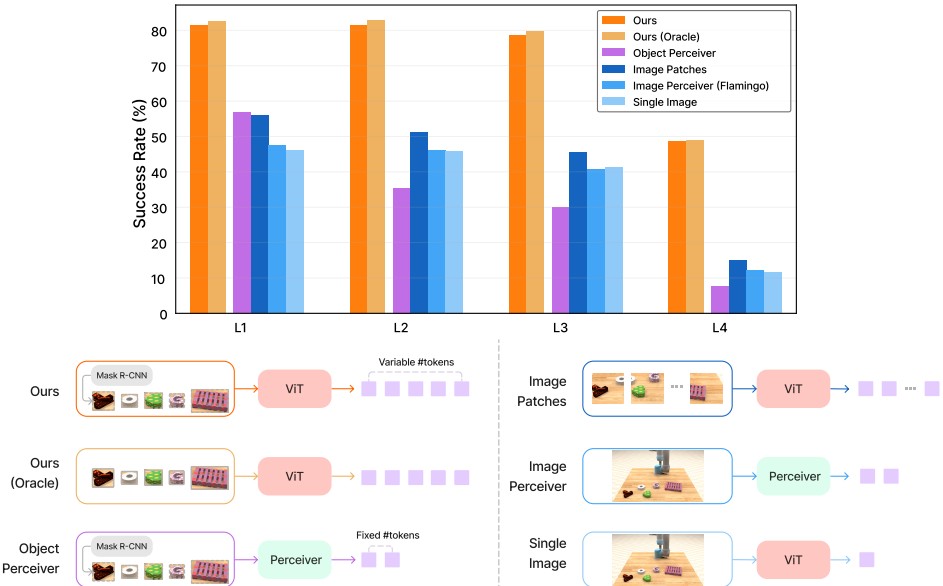

Figure 6: **Ablation on visual tokenizers**. We compare the performance of VIMA-200M model across different visual tokenizers. Our proposed object tokens outperform all methods that learn directly from raw pixels, and *Object Perceiver* that downsamples the object sequence to a fixed number of tokens.

in performance for VIMA is only *half* as severe as all other baselines. This results suggest that VIMA has developed more generalizable policy and robust representations than the competing approaches.

## 6.3 ABLATION STUDIES

Through extensive experiments, we ablate different design choices in VIMA and study their impact on robot decision making. We focus on the following 4 aspects: visual tokenization, prompt encoding, prompt conditioning variants, and robustness against distractors and imperfect prompts.

**Visual tokenization.** As explained in Sec. 5, VIMA processes the prompt and observation images into a variable number of object tokens with an off-the-shelf Mask R-CNN implementation. How important is this particular choice of visual tokenizer? We study 5 different variants and empirically evaluate their 4 levels of generalization performance on VIMA-BENCH. (1) **Ours (Oracle)**: instead of using Mask R-CNN, we directly read out the groundtruth bounding box from the simulator. In other words, we use a perfect object detector to estimate the upper bound on the performance of this study; (2) **Object Perceiver**: we apply a Perceiver module (Jaegle et al., 2021b;a) to convert the variable number of objects detected in each frame to a *fixed* number of tokens.

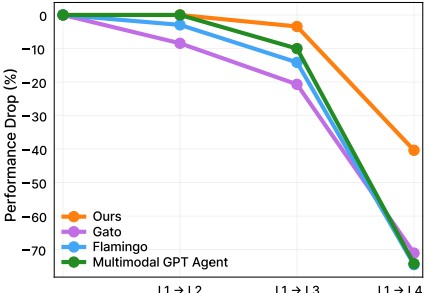

Figure 5: VIMA incurs much less performance drop than baselines as we evaluate on progressively harder zero-shot generalization.

Perceiver is more computationally efficient because it reduces the average sequence length; (3) **Image Perceiver**: the same architecture as the *Perceiver Resampler* in Flamingo, which converts an image to a small, fixed number of tokens; (4) **Image patches**: following Gato, we divide an RGB frame into square patches, and extract ViT embedding tokens. The number of patches is more than the output of Image Perceiver; (5) **Single image**: Decision Transformer's tokenizer, which encodes one image into a single token.

Fig. 6 shows the ablation results. We highlight a few findings. First, we note that our Mask R-CNN detection pipeline (Appendix, Sec. A.20) **incurs a minimal performance loss** compared to the oracle bounding boxes, thanks to the object augmentation (Sec. 5) that boosts robustness during training. Second, tokenizing from raw pixels (Image Perceiver, patches, or single embedding) consistently underperforms our object-centric format. We hypothesize that these tokenizers have to allocate extra

Figure 7: **Ablation: Prompt conditioning**. We compare our method (*xattn*: cross-attention prompt conditioning) with a vanilla transformer decoder (*gpt-decoder*) across different model sizes. Cross-attention is especially helpful in low-parameter regime and for harder generalization tasks.

internal capacity to parse the objects from low-level pixels, which likely impedes learning. Sax et al. (2018) echoes our finding that using mid-level vision can greatly improve agent generalization compared to an end-to-end pipeline. Third, even though *Ours* and *Object Perceiver* both use the same object bounding box inputs, the latter is significantly worse in decision making. We conclude that it is important to pass the **variable sequence of objects directly** to the robot controller rather than downsampling to a fixed number of tokens.

**Prompt Conditioning.** VIMA conditions the robot controller (decoder) on the encoded prompt by cross-attention. A simple alternative is to concatenate the prompt $\mathcal{P}$ and interaction history $\mathcal{H}$ into one big sequence, and then apply a decoder-only transformer like GPT (Radford et al., 2018) to predict actions. In this ablation, we keep the object tokenizer constant, and only switch the conditioning mechanism to causal sequence modeling. Note that this variant is conceptually "Gato with object tokens". Fig. 7 shows the comparison of VIMA (xattn) and the gpt-decoder variant across 4 generalization levels. While the variant achieves comparable performance in larger models, cross-attention still dominates in the small-capacity range and generalizes better in the most challenging L4 (*Novel Task*) setting. Our hypothesis is that cross-attention helps the controller stay better focused on the prompt instruction at each interaction step. This bears resemblance to the empirical results in Sanh et al. (2021); Wang et al. (2022b), which show that well-tuned encoder-decoder architectures can outperform GPT-3 in zero-shot generalization.

**Prompt Encoding.** We vary the size of the pre-trained T5 encoder to study the effect of prompt encoding. We experiment with three T5 capacities: small (30M), base (111M), to large (368M). For all T5 variants, we fine-tune the last two layers and freeze all other layers. We find no significant difference among the variants (Appendix, Sec. E.2), thus we set base as default for all our models.

**Policy Robustness.** We study the policy robustness against increased amounts of distractors and imperfect task specifications. See Appendix, Sec. E.3 for exact setup and results. VIMA exhibits minimal performance degradation with increased distractors and corrupted prompts. We attribute this robustness to the high-quality, pre-trained T5 language backbones.

# 7 CONCLUSION

Similar to GPT-3, a generalist robot agent should have an intuitive and expressive interface for human users to convey their intent. In this work, we introduce a novel *multimodal* prompting formulation that converts diverse robot manipulation tasks into a uniform sequence modeling problem. We propose VIMA, a conceptually simple transformer-based agent capable of solving tasks like visual goal, one-shot video imitation, and novel concept grounding with a single model. VIMA exhibits superior model and data scaling properties, and provides a strong starting point for future work.

The current VIMA experiments are not without limitations. We identify the following weaknesses: (1) limited action primitives (only pick-and-place and wipe for now); (2) limited simulator realism; (3) reliance on domain-finetuned Mask R-CNN to provide object tokens. However, VIMA's algorithm design is general-purpose and does not make assumptions about the particular observation and action formats. This opens the door to future works that may address many of these weaknesses with more sophisticated environments (e.g. BEHAVIOR (Srivastava et al., 2021)), stronger vision pipeline (large-scale open-vocabulary models like ViLD (Gu et al., 2021)), and temporally-extended robot controllers (such as MAPLE (Nasiriany et al., 2021)). With these stronger modules, VIMA could potentially scale to more challenging problems. We open-source all code to facilitate future research.

## 8 REPRODUCIBILITY STATEMENT

We provide comprehensive details to reproduce our work in the Appendix. Concretely, the specifications of each meta-task in the benchmarking suite are explained in Sec. B. Model architectures are elaborated in Sec. C. Hyperparameter configurations are listed in Sec. D. Furthermore, we host anonymized code at https://iclr3081.github.io/ for review.

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

## A    SIMULATOR DETAILS

We build our VIMA-BENCH simulation suite upon the Ravens physics simulator (Zeng et al., 2020; Shridhar et al., 2021). Specifically, it is supported by PyBullet (Coumans & Bai, 2016–2021) with a Universal Robot UR5 arm. The size of the tabletop workspace is $0.5 \times 1$m. Our benchmark contains extensible sets of object geometries and textures. Instantiated from an object-texture combination, all object instances can be rendered as RGB images appeared in multimodal prompts. Figure A.1 displays all object geometries. Figure A.2 displays all textures.

The observation space of VIMA-BENCH includes RGB images from both frontal and top-down views. It also includes a one-hot vector $\in \{0,1\}^2$ to indicate type of the end-effector $\in \{$suction cup, spatula$\}$. While a suction cup is equipped in most manipulation tasks, a spatula is used in particular for visual constraint tasks, where an agent is asked to "wipe" objects. VIMA-BENCH inherits the same action space from Zeng et al. (2020) and Shridhar et al. (2021), which consists of primitive actions of "pick and place" for tasks with a suction cup as the end effector, or "push" for tasks with a spatula. Both primitive actions contain two poses $\in \mathbf{SE}(2)$ specifying target poses of the end effector. For the "pick and place" primitive, they represent the pick pose and the place pose. Fir the "push" primitive, they represent the push starting pose and push ending pose.

Similar to prior work (Zeng et al., 2020; Shridhar et al., 2021), VIMA-BENCH provides scripted oracles to generate successful demonstrations for all tasks. We leverage them to construct an offline imitation dataset for behavioral cloning. Given a prompt, these pre-programmed bots can access privileged information such as the correct object to pick and target location to place.

## B    TASK SUITE

We develop 17 meta tasks that belong to 6 diverse categories. Thousands of individual tasks and their corresponding multimodal prompts can be procedually generated from these meta-task templates. We use PyBullet (Coumans & Bai, 2016–2021) as our backend and the default renderer to produce the RGB frames for training data and interactive test environments. For demonstration purpose, we apply the NVISII (Morrical et al., 2020) raytracing renderer to enhance the visual quality. We elaborate each meta task in the following subsections.

### B.1    SIMPLE OBJECT MANIPULATION

This task category asks agents to follow basic instructions specified by multimodal prompts.

**Task 01:**  Pick the specified object(s) and place it into the specified object.

- **Prompt**: `Put the {object}`$_1$ `into the {object}`$_2$.
- **Description**: The image placeholder $\{$`object`$\}_1$ is the object to be picked and the $\{$`object`$\}_2$ is the container object. The agent requires to recognize the objects with the correct color-shape combinations. To extend the difficulties, it supports more than one object to be picked or placed. For example, the prompt `Put the {object}`$_1$ `and {object}`$_2$ `into the {object}`$_3$. asks to pick two different objects and place into a target container. We uniformly sample different color-shape combos for objects to be picked and containers.
- **Success Criteria**: All specified object(s) to pick are within the bounds of the container object(s), with specified shapes and textures provided in the prompt.
- **Oracle Trajectory**: Shown in Fig. A.3 with its multimodal prompt.

**Task 02:**  In the workspace, put the objects with a specified texture shown in the scene image in the prompt into container object(s) with a specified color. This task requires the agent to find the correct object to manipulate by grounding the textural attributes from both natural language descriptions and the visual scene images.

- **Prompt**: `Put the {texture}`$_1$ `object in {scene} into the {texture}`$_2$ `object.`

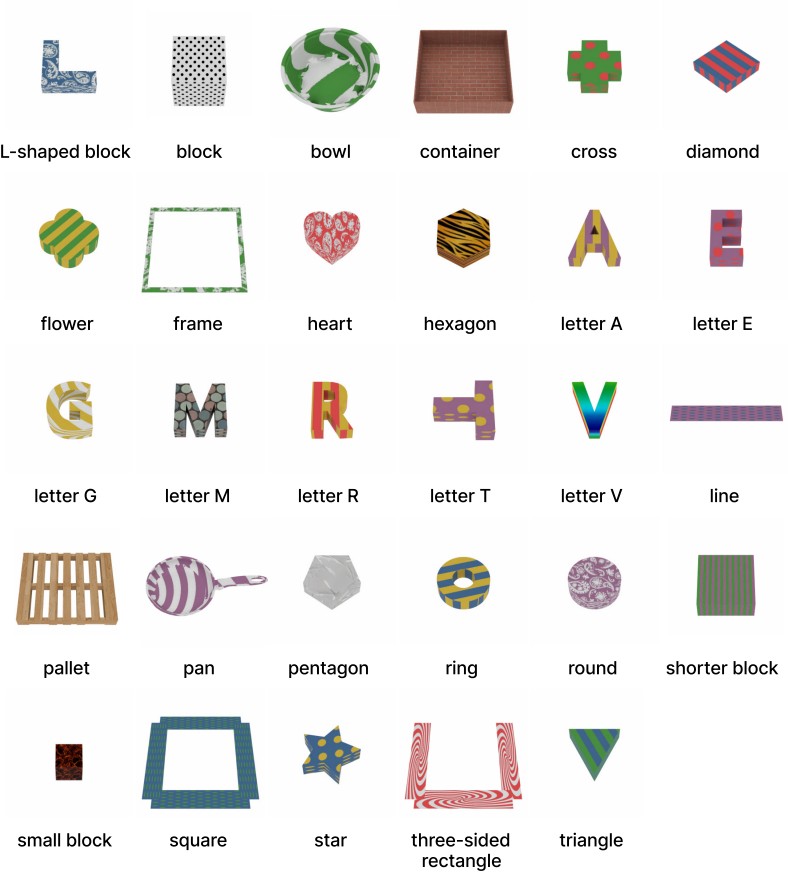

Figure A.1: **Object Gallery in VIMA-BENCH** textured with random textures. Bowl and pan are from Google Scanned Objects (Downs et al., 2022) while others are from Ravens (Zeng et al., 2020)

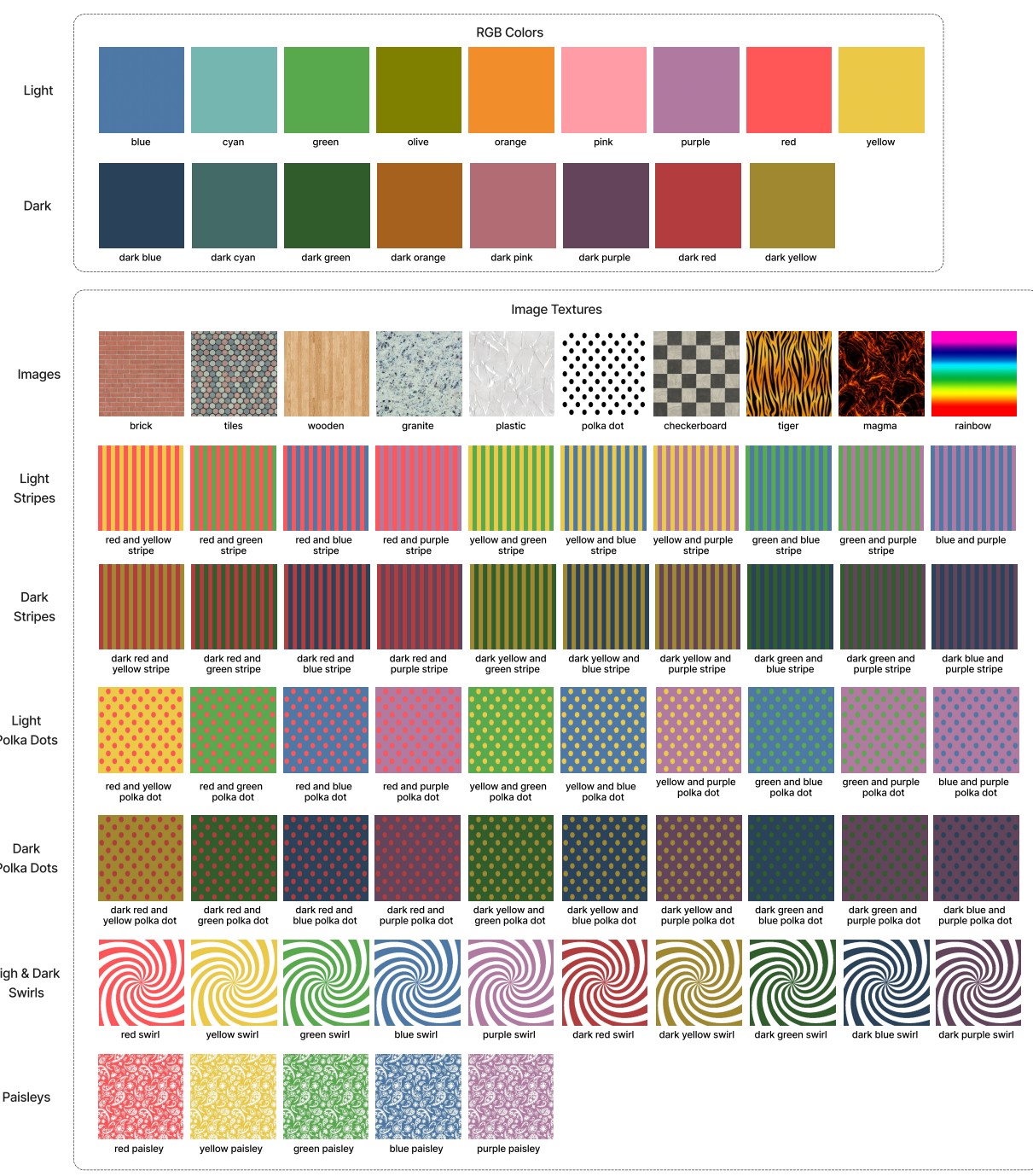

Figure A.2: **Texture Gallery in VIMA-BENCH**. The first row of image-based textures are from Blender Cloud Libraries (Weikert et al., 2022), while others are hard-coded.

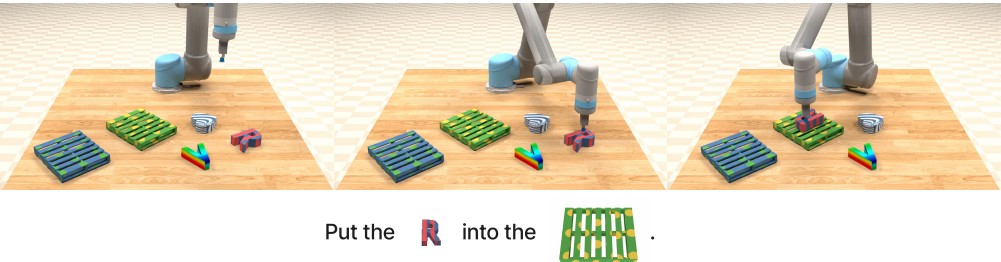

Figure A.3: Simple Object Manipulation: Task 01

- **Description**: The text placeholder $\{\texttt{texture}\}_1$ and $\{\texttt{texture}\}_2$ are sampled textures for objects to be picked and the container objects, respectively. The number of dragged objects with the same texture can be varied. $\{\texttt{scene}\}$ is the workspace-like image placeholder. There is a designated number of distractors with different textures (and potentially different shapes) in the scene. For each distractor in the workspace, it has $50\%$ chance to be either dragged or container distractor object with different textures from those specified in the prompt.
- **Success Criteria**: All objects in the workspace with $\{\texttt{texture}\}_1$ are within the bounds of the container object with $\{\texttt{texture}\}_2$.
- **Oracle Trajectory**: Shown in Fig. A.4 with its multimodal prompt.

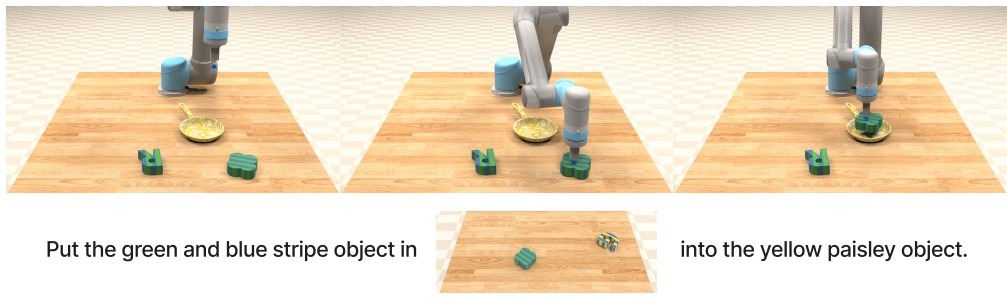

Figure A.4: Simple Object Manipulation: Task 02

**Task 03:** Rotate objects clockwise by certain degrees along $z$-axis. Only rotationally asymmetric objects are considered in this task.

- **Prompt**: `Rotate the` $\{\texttt{object}\}_1$ `{angles} degrees.`
- **Description**: The agent is required to rotate all objects in the workspace specified by the image placeholder $\{\texttt{object}\}_1$. There are also objects with different color-shape combinations in the workspace as distractors. $\{\texttt{angles}\}$ is the sampled degree that the dragged object needs to be rotated. A target angle is sampled from $30°$, $60°$, $90°$, $120°$, and $150°$.
- **Success Criteria**: The position of the specified object matches its original position, and the orientation matches the orientation after rotating specific angles.
- **Oracle Trajectory**: Shown in Fig. A.5 with its multimodal prompt.

## B.2 VISUAL GOAL REACHING

This task category requires agents to manipulate objects in the workspace to reach goal states represented as images shown in prompts.

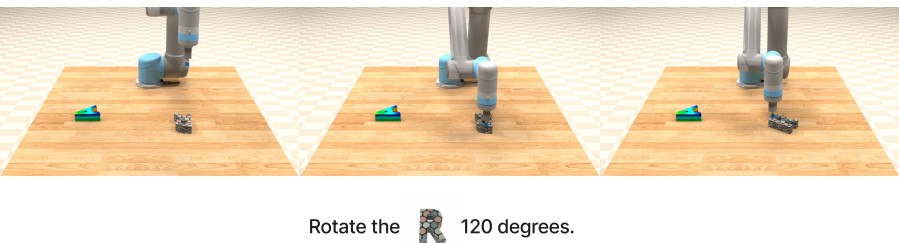

Rotate the 120 degrees.

Figure A.5: Simple Object Manipulation: Task 03

**Task 04:** Rearrange target objects in the workspace to match goal configuration shown in prompts. Note that to achieve the goal configuration, distractors may need to be moved away first.

- **Prompt**: `Rearrange to this {scene}.`

- **Description**: Objects in the scene placeholder {`scene`} are target objects to be manipulated and rearranged. In the workspace, the same target objects are spawned randomly, potentially with distractors randomly spawned as well. With a defined distractor conflict rate, the position of each distractor has this probability to occupy the position of any target object such that the rearrangement can only succeed if moving away that distractor first.

- **Success Criteria**: The configuration of target objects in the workspace matches that specified in the prompt.

- **Oracle Trajectory**: Shown in Fig. A.6 with its multimodal prompt. .

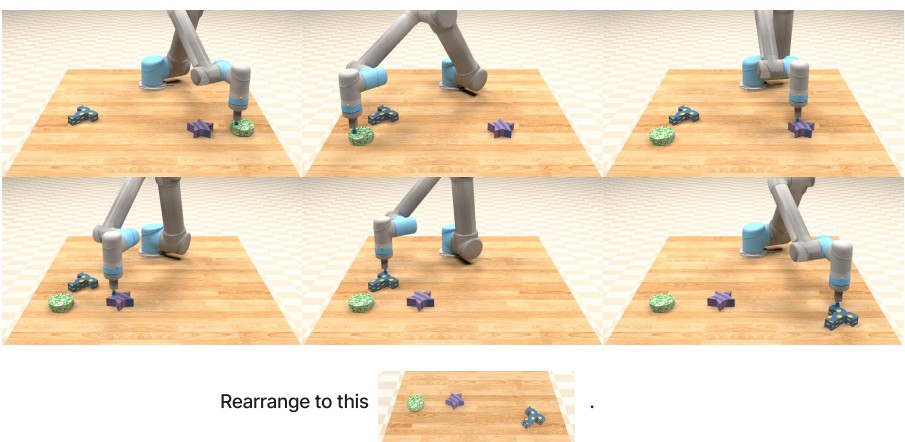

Rearrange to this .

Figure A.6: Visual Goal Reaching: Task 04

**Task 05:** Extend the task *04* by requiring the agent to restore rearranged objects to the initial setup after the "rearranging" phase.

- **Prompt**: `Rearrange objects to this setup {scene} and then restore.`

- **Description**: Same as the task *04*, except introducing the instruction "restore".

- **Success Criteria**: Meet the success criteria of the task *04*, and then within the allowed max steps restore all target objects to their initial configurations.

- **Oracle Trajectory**: Shown in Fig. A.7 with its multimodal prompt.

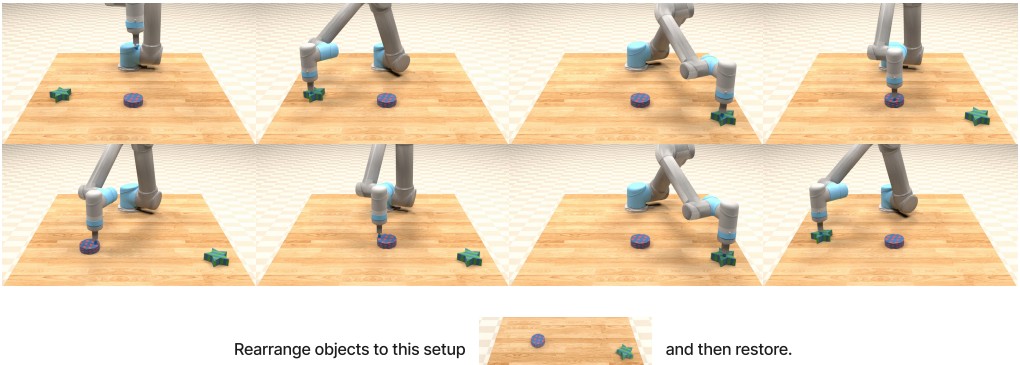

Rearrange objects to this setup       and then restore.

Figure A.7: Visual Goal Reaching: Task 05

### B.3   NOVEL CONCEPT GROUNDING

This task category requires agents to ground new concepts of adjectives, nouns, or verbs via visual perception and language understanding. Similar task design can be found in prior work (Hill et al., 2021). Completing these tasks are challenging, because the model should a) first understand prompts with interleaved texts, images, and even video frames; b) quickly internalize new concepts that are different across task instances, which even tests the ability to meta learn; and c) do complicated reasoning such as comparing between "taller" vs "less taller" vs "shorter" and then ground this reasoning into the robot action space.

Prompts consist of two parts: a definition part followed by an instruction part. In the definition part, novel conceptions are defined by multimodal illustrations with multiple support examples. In the instruction part, agents are asked to achieve the goal by properly applying concepts from the definition part. The assignment of unique nonsense words is varied and independent for each task instance such that tasks can only be solved if the agent applies the reasoning correctly. This ability is also referred to as *fast-mapping* (Heibeck & Markman, 1987).

**Task 06:**   Ground comparative adjectives by comparing the size or the textural saturation of objects and manipulating the correct object(s) instructed in the prompt.

- **Prompt**: `{demo_object}`$_1$ `is` `{novel_adj}` `than` `{demo_object}`$_2$`. Put the` `{adv}` `{novel_adj}` `{object}`$_1$ `into the` `{object}`$_2$`.`
- **Description**: The sampled adjective `{novel_adj}` is a dummy adjective placeholder for agent to ground. By default, the novel adjective set is `{daxer, blicker, modier, kobar}`. The real meaning can be related to size (smaller/larger) or textural saturation (lighter/darker texture). The image placeholders `{demo_object}`$_1$ and `{demo_object}`$_2$ illustrate how the novel adjective is defined. For example, if the real comparison is "taller", then the sampled object in `{demo_object}`$_1$ is taller than `{demo_object}`$_2$. The choices of the novel adjective and the real meaning are independently sampled for different task instances. For the instruction part, this task is similar to task *01*, where the agent is required to pick the specified dragged object(s) with the novel adjective attribute and then place it into the specified container object. To avoid revealing the correct object to manipulate, we use a neutral texture for objects appeared in the instruction part.
- **Success Criteria**: All target objects with the specified adjective attribute are within the bounds of the specified container object.
- **Oracle Trajectory**: Shown in Fig. A.8 with its multimodal prompt.

**Task 07:**   Orthogonal to task *06* by requiring to learn mappings of novel nouns.

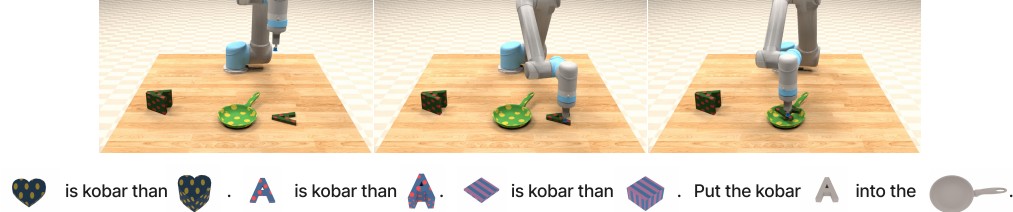

Figure A.8: Novel Concept Grounding: Task 06

- **Prompt**: `This is a {novel_name}_1 {object}_1.  This is a {novel_name}_2 {object}_2.  Put {novel_name}_1 into a {novel_name}_2.`
- **Description**: Novel noun words are defined with the text placeholders `{novel_name}_1` and `{novel_name}_2`, following their image placeholders `{object}_1` and `{object}_2`, for the target object and container object, respectively. Novel nouns are sampled from `{dax, blicket, wug, zup}`. In the instruction part, objects are expressed as novel nouns defined in the previous definition part. Distractors are defined the same as task *01*.
- **Success Criteria**: All target object(s) are within the bounds of the container object(s).
- **Oracle Trajectory**: Shown in Fig. A.8 with its multimodal prompt.

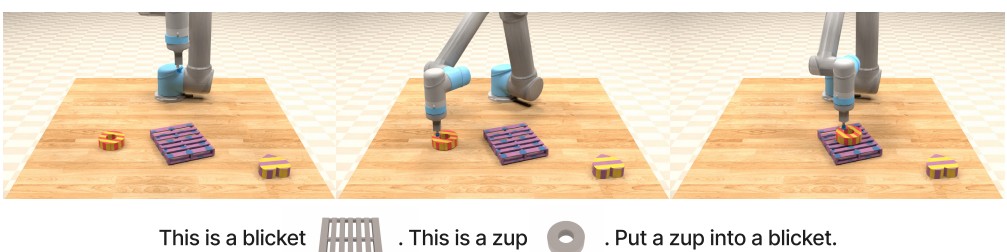

Figure A.9: Novel Concept Grounding: Task 07

**Task 08:**  Combination of tasks *06* and *07*.

- **Prompt**: `This is a {novel_name}_1 {object}_1.  This is a {novel_name}_2 {object}_2.  {demo_object}_1 is {adj} than {demo_object}_2. Put the {adv} {novel_adj} {novel_name}_1 into the {novel_name}_2.`
- **Description**: see task description for task *06* and task *07*.
- **Success Criteria**: Similar as tasks *06* and *07*.
- **Oracle Trajectory**: Shown in Fig. A.10 with its multimodal prompt.

**Task 09:**  A novel verb "twist" is defined as rotating a specific angle conveyed by several examples. This task is similar to task *03*, but it requires the agent to infer what is the exact angle to rotate from the prompt and to ground novel verbs that are semantically similar but different in exact definitions.

- **Prompt**: `"Twist" is defined as rotating object a specific angle. For examples:  From {before_twist}_i to {after_twist}_i. Now twist all {texture} objects.`
- **Description**: Both `{before_twist}_i` and `{after_twist}_i` are scene placeholders where `{before_twist}_i` shows a randomly sampled object before "twist" and `{after_twist}_i` shows the same object pose after "twist". All examples illustrate the

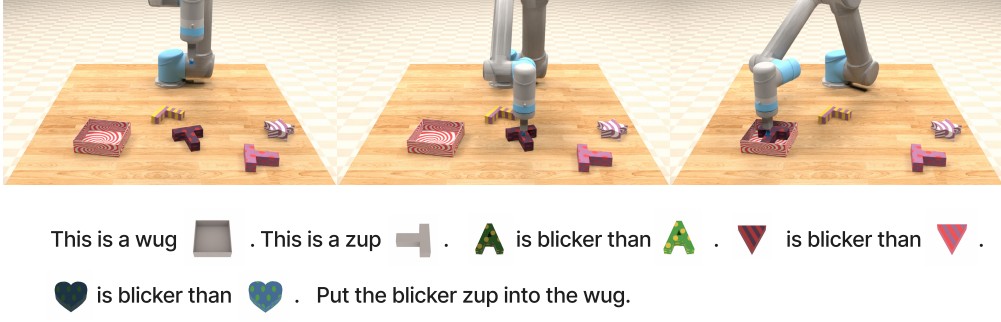

Figure A.10: Novel Concept Grounding: Task 08

same sampled angle of the rotation. In the workspace, the target objects have the texture specified by {texture} and randomly sampled shapes.

- **Success Criteria**: Same as the task *03*.
- **Oracle Trajectory**: Shown in Fig. A.11 with its multimodal prompt.

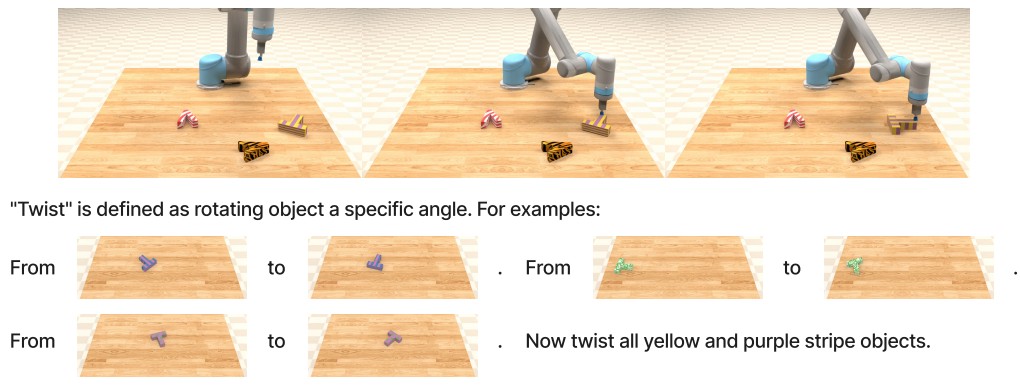

Figure A.11: Novel Concept Grounding: Task 09

## B.4    ONE-SHOT VIDEO IMITATION

This task category requires agents to imitate motions demonstrated through videos shown in prompts. We follow prior works (Finn et al., 2017; Dasari & Gupta, 2020; Duan et al., 2017) to formulate the problem by giving one video demonstration (represented as consecutive frames in prompts), then test the learned imitator's ability to produce target trajectories. This setup is challenging because a) only one demonstration is available to the agent; b) the model needs to understand video frames interleaved with textual instructions; and c) missing correspondences between demonstrations and target trajectories since demonstrations only show partial key frames.

**Task 10:**    Follow motions for specific objects.

- **Prompt**: `Follow this motion for {object}: {frame}`$_1$`...{frame}`$_i$`... {frame}`$_n$`.`
- **Description**: Image placeholder {object} is the target object to be manipulated and {{frame}$_i$} is set of workspace-like scene placeholders to represent a video trajectory, where $n$ is the trajectory length. There is an object spawned at the center in both the

workspace and the prompt video but with different textures as a distractor. The initial position of the target object matches that in $\{\texttt{frame}\}_1$.

- **Success Criteria**: In each step, the pose of the target object matches the pose in the corresponding video frame. Incorrect manipulation sequences are considered as failures.
- **Oracle Trajectory**: Shown in Fig. A.12 with its multimodal prompt.

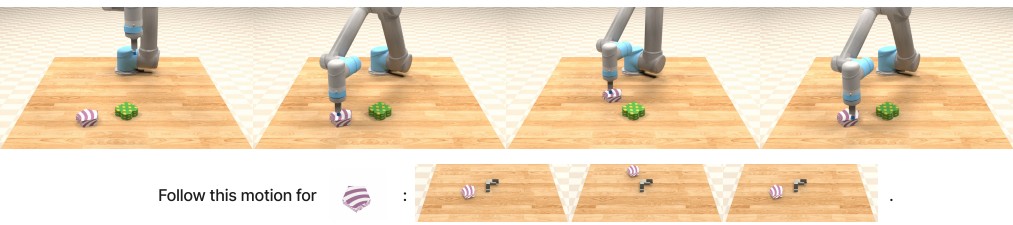

Figure A.12: One-shot video imitation: Task 10

**Task 11:** Stack objects with the order illustrated in the prompt video.

- **Prompt**: `Stack objects in this order` $\{\texttt{frame}\}_1 \ldots \{\texttt{frame}\}_i \ldots \{\texttt{frame}\}_n$.
- **Description**: There are multiple objects with the same shape but different textures spawned in the workspace without any stacking initially. Distractor objects with different shapes are spawned in the workspace but not in the prompt video. At each step of the prompt video, one of the top objects is stacked over another object or put at an empty position.
- **Success Criteria**: Similar as task 10.
- **Oracle Trajectory**: Shown in Fig. A.13 with its multimodal prompt.

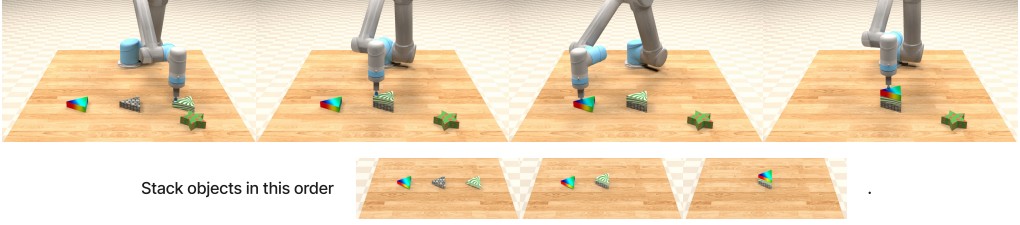

Figure A.13: One-shot video imitation: Task 11

### B.5 VISUAL CONSTRAINT SATISFACTION

This task category requires agents to wipe a specific number of objects in the workspace to a goal region while also satisfy the given visual constraint.

**Task 12:** Sweep the designated number of objects into a specified region without exceeding the boundary.

- **Prompt**: `Sweep {quantifier} {object} into {bounds} without exceeding {constraint}.`
- **Description**: `{object}` is the image placeholder of the target object to be swept spawned with a random amount in the workspace. Distractors have the same amount, same shape, but different color from target objects. `{quantifier}` is the text placeholder to determine the target quantity of objects to be wiped, sampled from `any`, `one`, `two`, `three`, and `all`. `{bounds}` is the image placeholder for a three-sided rectangle as the goal region. `{constraint}` is the constraint line.

- **Success Criteria**: The exact number of target objects to be swept are all inside the specified region. Failure reasons include 1) any distractor being wiped into the region, 2) target object exceeding the constraint, or 3) incorrect number of target objects being swept into the goal region.
- **Oracle Trajectory**: Shown in Fig. A.14 with its multimodal prompt.

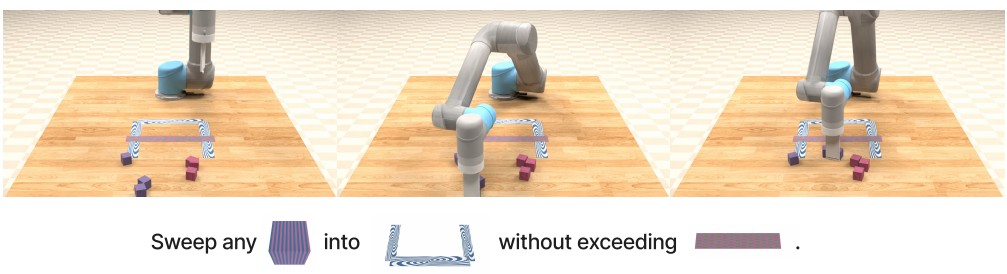

Figure A.14: Visual Constraint Satisfaction: Task 12

**Task 13:** Sweep the designated number of objects into a specified region without touching the constraint.

- **Prompt**: `Sweep {quantifier} {object} into {bounds} without touching {constraint}.`
- **Description**: Similar as task *12* but requiring different way to satisfy the constraint. The agent has to learn to avoid "touching" the constraint line in this case.
- **Success Criteria**: Similar as task *12* except that the constraint is to not touch the red line.
- **Oracle Trajectory**: Shown in Fig. A.15 with its multimodal prompt.

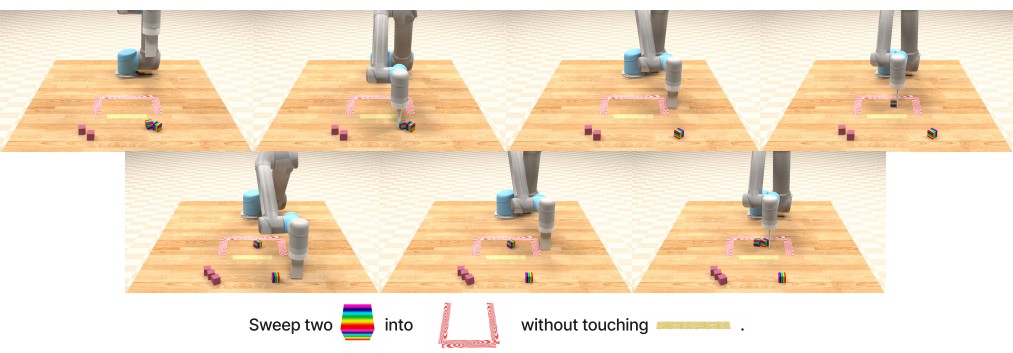

Figure A.15: Visual Constraint Satisfaction: Task 13

## B.6 VISUAL REASONING

This task category requires agents to make decisions by reasoning over or memorizing information conveyed through multimodal prompts.

**Task 14:** By reasoning the "same texture", the agent is required to pick all objects in the workspace with the same texture as the container objects specified in the prompt and place them into it.

- **Prompt**: `Put all objects with the same texture as {object} into it.`

- **Description**: {object} is the sampled goal container object. In the workspace, there are objects with the same texture as the container but potentially different shapes. Distractors with different textures are spawned.
- **Success Criteria**: All objects with the same texture as the goal container are within the bounds of the container.
- **Oracle Trajectory**: Shown in Fig. A.16 with its multimodal prompt.

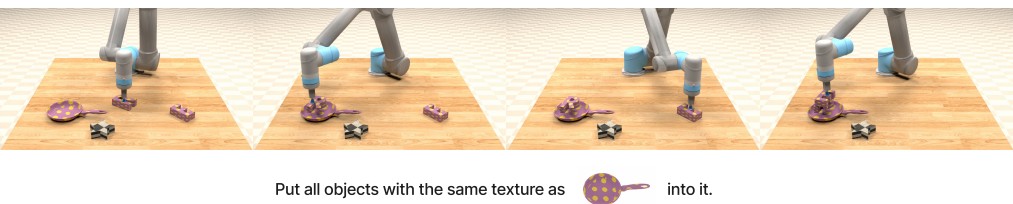

Put all objects with the same texture as into it.

Figure A.16: Visual Reasoning: Task 14

**Task 15:** By reasoning the "same shape", the agent is required to pick all objects in the workspace with the same top-down shape as the goal container specified in the prompt and place them into it. For example, blocks and boxes have the same rectangular shape.

- **Prompt**: `Put all objects with the same profile as {object} into it.`
- **Description**: Similar to the task *14* except the objects to be picked and placed are with the same shape. There are three different shapes: *rectangular-like* (e.g. block and pallet), *circle-like* (e.g. ring and bowl), and *undetermined* for the rest.
- **Success Criteria**: All objects with the same shape as the container are within the container.
- **Oracle Trajectory**: Shown in Fig. A.17 with its multimodal prompt.

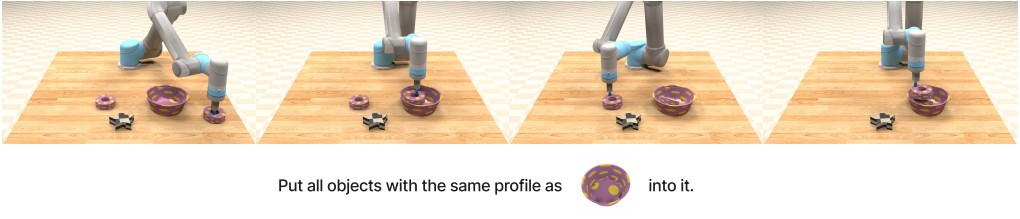

Put all objects with the same profile as into it.

Figure A.17: Visual Reasoning: Task 15

**Task 16:** Put the target object into the container, and then put one of its old neighbors into the same container.

- **Prompt**: `First put {object}₁ into {object}₂ then put the object that was previously at its {direction} into the same {object}₂.`
- **Description**: Objects in image placeholders {object}₁ and {object}₂ are the target object to be picked and the container, respectively. We then ask the agent to put one of old neighbors of the previous target object into the same container. The old neighboring object is specified through cardinal directions {north, south, west, east}.
- **Success Criteria**: The target object and the correct neighboring object are inside the container.
- **Oracle Trajectory**: Shown in Fig. A.18 with its multimodal prompt.

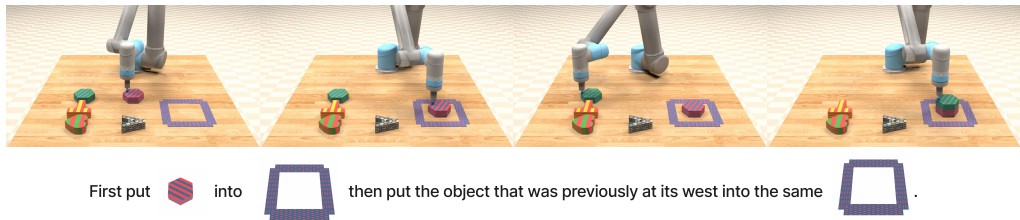

Figure A.18: Visual Reasoning: Task 16

**Task 17:** Pick and place the target object specified in the prompt into different containers in order then restore to the initial container.

- **Prompt**: `Put {object}`$_1$ `into {object}`$_2$`. Finally restore it into its original container.`

- **Description**: The object in the image placeholder `{object}`$_1$ is the target object to be manipulated across the task. There are more than one target containers (e.g. `Put {object}`$_1$ `into {object}`$_2$ `then {object}`$_3$`.Finally restore it into its original container.` for two target base objects to be placed in order). The rest of spawned containers naturally becomes distractors.

- **Success Criteria**: The target object are first put into multiple containers following the specific order. Finally it should be restored into its original container.

- **Oracle Trajectory**: Shown in Fig.A.19 with its multimodal prompt.

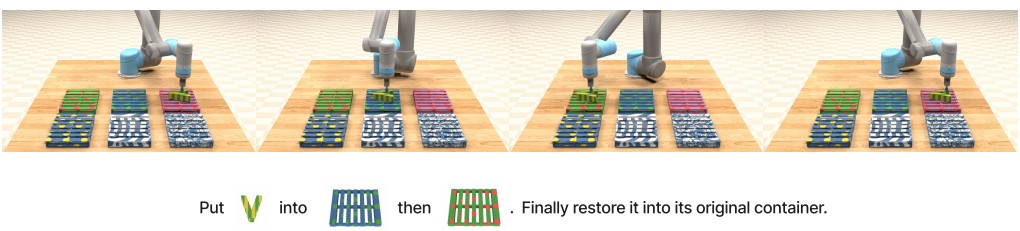

Figure A.19: Visual Reasoning: Task 17

## C  MODEL ARCHITECTURE

In this section, we provide comprehensive details about VIMA model architecture as well as other adapted baseline methods. We implement all models in PyTorch (Paszke et al., 2019) and adapt Transformer-related implementation from Wolf et al. (2019).

### C.1  SUMMARY OF DIFFERENT METHODS

We summarizes differences between VIMA and other baseline methods in Table 1. In the column "Prompt Conditioning", an alternative of cross-attention is to first concatenate prompt and interaction into a big sequence, then repetitively apply transformer decoders to predict actions. It is referred to as "direct modeling". The relative computation cost is quadratically proportional to number of observation tokens.

Table 1: Comparison of different methods.

|  | Visual Tokenizer | Prompt Conditioning | Number of Observation Tokens per Step |
|---|---|---|---|
| Ours | Object tokens consisting of cropped images and bounding boxes | Cross-attention | Equal to number of objects, typically 3 to 8 |
| Gato (Reed et al., 2022) | Image patch tokens encoded by a ViT | Direct modeling | Equal to number of image patches, 16 |
| Flamingo Agent (Alayrac et al., 2022) | Image patch tokens encoded by a ViT, further downsampled by a Perceiver module | Cross-attention | Equal to number of learned query vectors, 4 |
| Multimodal GPT Agent (Brown et al., 2020) | Single image token encoded by a ViT | Direct modeling | Single visual feature, 1 |

Table 2: Model hyperparameters for multimodal prompt tokenization.

| Hyperparameter | Value |
|---|---|
| **Text Tokenization** | |
| Tokenizer | `t5-base` tokenizer |
| Embedding Dimension | 768 |
| **Image Tokenization** | |
| ViT Input Image Size | $32 \times 32$ |
| ViT Patch Size | 16 |
| ViT Width | 768 |
| ViT Layer | 4 |
| ViT Number of Heads | 24 |
| **Bounding Box MLP** | |
| Hidden Dimension | 768 |
| Hidden Depth | 2 |
| **Prompt Encoding** | |
| Pre-trained LM | `t5-base` |
| Unfreeze Last N Layers | 2 |
| Positional Embedding | Absolute |
| Token Adapter MLP Depth | 2 |

## C.2 VIMA ARCHITECTURE

### C.2.1 MULTIMODAL PROMPT TOKENIZATIONS

As introduced in Section 5, there are 3 types of input formats in multimodal prompts, namely (1) **text inputs**, (2) **images of full scenes**, and (3) **images of single objects**.

For **text inputs**, we follow the standard pipeline in NLP to first tokenize raw languages to discrete indices through pre-trained `t5-base` tokenizer. We then obtain corresponding word tokens from the embedding look-up of the pre-trained `t5-base` model. For **images of full scenes**, we first parse the scene through a fine-tuned mask R-CNN detection model (He et al., 2017; Wu et al., 2019) to extract individual objects. Each object representation contains a bounding box and a cropped image. The bounding box is in the format of $[x_{\text{center}}, y_{\text{center}}, \text{height}, \text{width}]$. We normalize it to be within $[0, 1]$ by dividing each dimension with corresponding upper-bound value. We then pass it through a bounding box encoder MLP and obtain a feature vector. To process the cropped image, we first pad non-square image to a square by padding along the shorter dimension. We then resize it to a pre-configured size and pass it through a ViT (trained from scratch) to obtain the image feature. Finally, an object token is obtained by concatenating the bounding box feature and the image feature and mapping to the embedding dimension. For **images of single objects**, we obtain tokens in the same way except with a dummy bounding box. Detailed model hyperparameters about tokenizations are listed in Table 2.

After obtaining a sequence of prompt tokens, we follow Tsimpoukelli et al. (2021) to pass it through a pre-trained `t5-base` encoder to obtain encoded prompt. Note that we add adapter MLP between object tokens and the T5 encoder. We adopt learned absolute positional embedding. Model hyperparameters are listed in Table 2 as well.

Table 3: Model hyperparameters for observation encoding.

| Hyperparameter | Value |
| --- | --- |
| Observation Token Dimension | 768 |
| End Effector Embedding Dimension | 2 |
| Positional Embedding | Absolute |

Table 4: Model hyperparameters for action decoders.

| Hyperparameter | Value |
| --- | --- |
| Hidden Dimension | 512 |
| Hidden Depth | 2 |
| Activation | ReLU |
| X-axis Discrete Bins | 50 |
| Y-axis Discrete Bins | 100 |
| Rotation Discrete Bins | 50 |

### C.2.2 OBSERVATION ENCODING

Since all RGB observations are images of full scenes, we follow the same procedure discussed above to obtain flattened object tokens. Because we provide RGBs from two views (frontal and top-down), we order object tokens by following the order of [frontal, top-down]. We one-hot encode the state of the end effector. We then concatenate object tokens with the end-effector state and transform to observation tokens. We adopt learned absolute positional embedding. Detailed model hyperparameters about observation encoding is provided in Table 3.

### C.2.3 ACTION ENCODING

Since our model is conditioned on observation-action interleaved history, we also tokenize past actions. We follow common practice in Chen et al. (2021); Zheng et al. (2022) to encode past actions with a two-layer MLP. It has a hidden dimension of 256. We then map outputs to token dimension and obtain action tokens.

### C.2.4 SEQUENCE MODELING

The robot controller in VIMA is a causal decoder that autoregressively predicts actions. To condition the decoder on prompt tokens, we perform cross-attention between history tokens and prompt tokens (Figure 3). Concretely, we pass history tokens as the query sequence and prompt tokens as the key-value sequence into cross-attention blocks. The output prompt-aware trajectory tokens then go through causal self-attention blocks. We alternate cross-attention and self-attention $L$ times. This procedure is technically described in Pseudocode 1.

Table 5: Model hyperparameters for ViT used in baseline methods.

| Hyperparameter | Value |
| --- | --- |
| Image Size | 64 x 128 |
| Patch Size | 32 |
| ViT Width | 768 |
| ViT Layers | 4 |
| ViT Heads | 24 |

Table 6: Model hyperparameters for Perceiver Resampler used in Flamingo method.

| Hyperparameter | Value |
| --- | --- |
| Number of Latent Queries | 4 |
| Number of Blocks | 4 |
| Self-attn per Block | 4 |
| Self-attn Heads | 24 |
| Cross-attn Heads | 24 |

```python
def xattn_sequence_modeling(
    prompt_tokens,      # the [L, d] prompt tokens (L=prompt length)
    obs_tokens,         # the [T, d] obs tokens (T=time step)
    act_tokens,         # the [T-1, d] action tokens
    traj_pos_embd,      # learned positional embedding for trajectory
    prompt_pos_embd,    # learned positional embedding for prompt
):
    # interleave obs and action tokens
    traj_tokens = interleave(obs_tokens, act_tokens)   # [2T-1, d]
    # add positional embedding to trajectory tokens
    x = traj_tokens + traj_pos_embd
    # add positional embedding to prompt tokens
    prompt_tokens = prompt_tokens + prompt_pos_embd

    # apply xattn and causal self-attn
    for i in range(num_layers):
        # cross-attention
        x = x + attn_i(q=x, kv=prompt_tokens)
        # feed forward
        x = x + ffw_xattn_i(x)
        # self-attention
        x = x + causal_attn_i(q=x, kv=x)
        # feed forward
        x = x + ffw_i(x)

    # the last token is the predicted action token
    predicted_act_token = x[-1]
    return predicted_act_token
```

Pseudocode 1: Cross-attention operation that conditions the trajectory history on prompt. We repetitively alternate cross-attention and self-attention to model the trajectory given a specific task.

### C.2.5   ACTION DECODING

After obtaining the predicted action token, we map it to the action space $\mathcal{A}$ and obtain the predicted action. This is achieved though a group of action heads. Since the action space consists of two $\mathbf{SE}(2)$ poses, for each pose we use six independent heads to decode discrete actions (two for xy coordinate and four for rotation represented in quaternion). These discrete actions are then de-discretized and mapped to continuous actions through affine transformation. The two poses are modeled independently. Early ablations show that this independent modeling is equally good as alternatives techniques like autoregressive decoding (Vinyals et al., 2019; OpenAI et al., 2019). Detailed model hyperparameters are listed in Table 4.

### C.3   BASELINES ARCHITECTURES

In this section, we elaborate model architectures for baseline methods. Some components such as the action decoder are same across all baseline methods and ours. Therefore, we only discuss unique model components.

### C.3.1 GATO

**Gato** (Reed et al., 2022) introduces a decoder-only model that solves tasks from multiple domains including robotics, video game, image captioning, language modeling, etc. Different tasks are specified by supplying the model with an initial sequence of corresponding tokens. For example, in tasks involving decision making, these tokens include observation and action tokens. For fair comparison, we provide the same conditioning as VIMA, i.e., our multimodal tokenized prompts. Similar to our method, Gato also predicts actions in an autoregressive manner. Gato and our method share the same training philosophy to only optimize the causal behavior cloning objective. However, unlike our method that adopts an object-centric representation to treat individual objects as observation tokens, Gato divides input images into patches and encodes them by a ViT (Dosovitskiy et al., 2020) model to produce observation tokens. Furthermore, Gato relies on causal self-attention to model entire trajectory sequences starting with prompt tokens. Hyperparameters of Gato's ViT is listed in Table 5. The transformer-decoder style sequence modeling is technically illustrated in Pseudocode 2.

```
def causal_sequence_modeling(
    prompt_tokens,   # the [L, d] prompt tokens (L=prompt length)
    sep_token,       # the [1, d] learned token to separate prompt and
    trajectory history
    obs_tokens,      # the [T, d] obs tokens (T=time step)
    act_tokens,      # the [T-1, d] action tokens
    pos_embd,        # learned positional embedding
):
    # interleave obs and action tokens
    traj_tokens = interleave(obs_tokens, act_tokens)  # [2T-1, d]
    # assemble input tokens
    x = concat([prompt_tokens, sep_token, traj_tokens])
    x = x + pos_embd

    # apply GPT layers with causal mask
    for i in range(num_layers):
        # self-attention
        x = x + causal_attn_i(q=x, kv=x)
        # feed forward
        x = x + ffw_i(x)

    # the last token is the predicted action token
    predicted_act_token = x[-1]
    return predicted_act_token
```

Pseudocode 2: Plain sequence modeling that temporally concatenates prompt and trajectory history and repetitively perform causal self-attention operation.

### C.3.2 FLAMINGO

**Flamingo** (Alayrac et al., 2022) is a vision-language model that learns to generate textual completion in response to multimodal prompts. It embeds a variable number of prompt images into a fixed number of tokens via the Perceiver Resampler module (Jaegle et al., 2021b), and conditions the language decoder on encoded prompts by cross-attention. Flamingo does not work with embodied agents out of the box. We adapt it by replacing the output layer with robot action heads (hyperparameters listed in Table 4) and using tokenized rollout histories as inputs. We train it end-to-end with causal behavior cloning loss. The modified Flamingo agent differs from ours since it processes image observations into a fixed number of visual tokens through a learned Perceiver Resampler. Model hyperparameters for our reimplementation of the Perceiver Resampler is listed in Table 6.

### C.3.3 MULTIMODAL GPT AGENT

**Multimodal GPT agent** (Brown et al., 2020) is a behavior cloning agent conditioned on tokenized multimodal prompts with the GPT architecture. It autoregressively decodes next actions given multimodal prompts and interaction histories. We optimize this method end-to-end with causal behavior cloning loss. Similar to prior works of casting RL problems as sequence modeling (Chen et al., 2021; Janner et al., 2021; Zheng et al., 2022), it encodes an image into a single "state" token through a learned ViT encoder. It also directly models entire trajectory sequences prepended with prompt tokens. Therefore, it differs from our method in the representation of observation tokens and prompt conditioning. For visual tokenizer, we employ a learned ViT with hyperparameters listed in Table 5.

### C.4 MASK R-CNN DETECTION MODEL

Finally, we elaborate the mask R-CNN model (He et al., 2017) for scene parsing and object extraction. We fine-tuned a pre-trained lightweight mask R-CNN (mask_rcnn_R_50_FPN_3x) from Wu et al. (2019) to adapt to scenes and images in our tabletop environment. A visualization of its output is provided in Figure A.20. We do not use the predicted object names in our models.

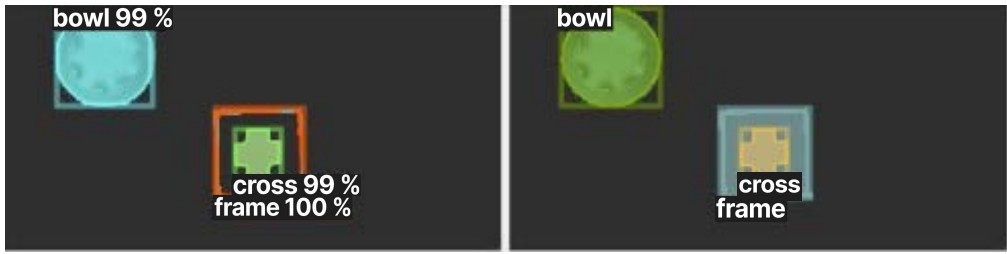

Figure A.20: Visualization of fine-tuned mask R-CNN. Left: Prediction from the detection model. Right: Ground-truth scene parsing. The detection model agrees well with ground-truth objects.

## D   VIMA TRAINING DETAILS

We follow the best practices to train Transformer models using the AdamW optimizer (Loshchilov & Hutter, 2019), learning rate warm-up, cosine annealing (Loshchilov & Hutter, 2017), etc. Training hyperparameters are provided in Table 7. We use GEGLU activation (Shazeer, 2020) inside Transformer models across all methods.

Table 7: Hyperparameters used during training.

| Hyperparameter | Value |
| --- | --- |
| Learning Rate | 0.0001 |
| Warmup Steps | 7K |
| LR Cosine Annealing Steps | 17K |
| Weight Decay | 0 |
| Dropout | 0.1 |
| Gradient Clip Threshold | 1.0 |

To make trained models robust to detection inaccuracies and failures, we apply *object augmentation* by randomly injecting *false-positive* detection outputs. Concretely, for observation at each time step, we sample number of augmented objects i.i.d. $n_{\text{augmented objects}} \sim \text{Cat}(K, \mathbf{p})$, where $\text{Cat}(\cdot)$ denotes a multi-categorical distribution with $K$ supports parameterized by $\mathbf{p}$. For each augmented object, we then randomly sample a bounding box and corresponding cropped image to add to object tokens. In our experiments, we set $\mathbf{p} = \{0 : 0.95, 1 : 0.05\}$ with $K = 2$.

## D.1 VARY MODEL CAPACITY

We train a spectrum of 7 models ranging from 2M to 200M parameters. To vary the model capacity, we follow prior work (Chowdhery et al., 2022) to change embedding dimension and number of layers. We list configurations for methods with cross-attention prompt conditioning (i.e., ours and Flamingo) in Table 8, and configurations for methods only with causal self-attention (i.e., Gato and DT) in Table 9.

Table 8: Configurations for different sized models with cross-attention prompt conditioning.

| Model Size (M) | Embedding Dimension | Num Blocks | X-attn Heads | Self-attn Heads |
|---|---|---|---|---|
| 2 | 256 | 1 | 8 | 8 |
| 4 | 256 | 2 | 8 | 8 |
| 9 | 320 | 3 | 10 | 10 |
| 20 | 384 | 4 | 12 | 12 |
| 43 | 512 | 5 | 16 | 16 |
| 92 | 640 | 7 | 20 | 20 |
| 200 | 768 | 11 | 24 | 24 |

Table 9: Configurations for different sized models with causal self-attention prompt conditioning.

| Model Size (M) | Embedding Dimension | Num Blocks | Self-attn Heads |
|---|---|---|---|
| 2 | 64 | 1 | 2 |
| 4 | 96 | 2 | 3 |
| 9 | 192 | 3 | 6 |
| 20 | 320 | 4 | 10 |
| 43 | 512 | 5 | 16 |
| 92 | 768 | 7 | 24 |
| 200 | 768 | 18 | 24 |

## E  MORE EXPERIMENT RESULTS

### E.1  BREAKDOWN RESULTS

We show breakdown results for Figure 4 in Tables 10, 11, 12, and 13, respectively.

### E.2  VARY T5 ENCODER SIZES

We vary the size of the pre-trained T5 encoder (Raffel et al., 2020) to study the effect of prompt encoding. We experiment with three T5 model capacities: t5-small (30M), t5-base (111M), to t5-large (368M). For all T5 variants, we fine-tune the last two layers and freeze all other layers. We fix the parameter count of the decision-making part to be 200M. As shown in Table 14, we find no significant difference among the variants. Thus we set the standard t5-base as default for all our models.

### E.3  POLICY ROBUSTNESS

**Increased Amounts of Distractors.** We study the policy robustness against increased amounts of distractors in scenes. For all tasks being evaluated, we add one more distractor object. We ran our largest VIMA model with 200M parameters. The result is presented in Table 15.

It turns out that the performance of VIMA degrades minimally with more distractors than the training distribution. This indicates that our agent has learned a reasonably robust policy against objects that are irrelevant to the task.

Table 10: L1 level generalization results. Model indicates robot controller parameter count.

| Model | Method | Task 01 | Task 02 | Task 03 | Task 04 | Task 05 | Task 06 | Task 07 | Task 09 | Task 11 | Task 12 | Task 15 | Task 16 | Task 17 |
|---|---|---|---|---|---|---|---|---|---|---|---|---|---|---|
| 2M | Ours | **100.0** | **100.0** | **100.0** | **96.0** | 37.0 | **100.0** | **100.0** | **9.5** | **87.0** | 64.0 | **93.5** | **45.0** | **63.0** |
| | Gato | 62.0 | 61.0 | 22.5 | 13.5 | 7.0 | 44.5 | 54.0 | 4.0 | 48.0 | **85.0** | 44.5 | 43.0 | 0.0 |
| | Flamingo | 56.0 | 56.0 | 53.5 | 36.5 | **37.5** | 45.0 | 55.5 | 3.5 | 54.0 | 83.5 | 40.5 | 28.5 | 2.0 |
| | Multimodal GPT | 59.5 | 50.5 | 7.5 | 7.0 | 0.5 | 43.5 | 49.5 | 2.0 | 61.5 | 76.5 | 27.5 | 5.0 | 0.0 |
| 20M | Ours | **100.0** | **100.0** | **100.0** | **99.5** | **59.5** | **100.0** | **100.0** | **13.5** | 74.0 | 72.5 | **96.5** | 39.5 | **47.5** |
| | Gato | 61.5 | 62.0 | 32.5 | 49.0 | 38.0 | 46.0 | 60.0 | 5.0 | 68.0 | 83.0 | 47.0 | **46.5** | 2.0 |
| | Flamingo | 63.0 | 61.5 | 55.0 | 50.0 | 42.5 | 41.5 | 58.0 | 6.0 | 62.0 | 83.0 | 44.0 | 38.5 | 1.0 |
| | Multimodal GPT | 60.5 | 64.0 | 50.5 | 44.0 | 41.0 | 48.0 | 61.5 | 7.0 | **85.0** | **84.0** | 44.5 | 39.0 | 2.5 |
| 200M | Ours | **100.0** | **100.0** | 99.5 | **100.0** | 56.5 | **100.0** | **100.0** | 18.0 | **77.0** | 93.0 | **97.0** | 76.5 | **43.0** |
| | Gato | 79.0 | 68.0 | 91.5 | 57.0 | 44.5 | 54.0 | 74.0 | 18.0 | 61.0 | 88.5 | 83.5 | 33.5 | 2.5 |
| | Flamingo | 56.0 | 58.5 | 63.0 | 48.5 | 38.0 | 48.5 | 62.5 | 3.5 | 66.5 | 86.0 | 40.0 | 43.5 | 2.5 |
| | Multimodal GPT | 62.0 | 57.5 | 41.0 | 55.5 | 45.5 | 47.5 | 54.5 | 8.5 | **77.0** | 81.5 | 41.0 | 38.0 | 0.5 |

Table 11: L2 level generalization results. Model indicates robot controller parameter count.

| Model | Method | Task 01 | Task 02 | Task 03 | Task 04 | Task 05 | Task 06 | Task 07 | Task 09 | Task 11 | Task 12 | Task 15 | Task 16 | Task 17 |
|---|---|---|---|---|---|---|---|---|---|---|---|---|---|---|
| 2M | Ours | **100.0** | **100.0** | **100.0** | **95.5** | **37.5** | **100.0** | **100.0** | **17.5** | **87.5** | 67.0 | **97.5** | **46.0** | **54.5** |
| | Gato | 49.5 | 49.0 | 23.0 | 17.5 | 0.5 | 47.5 | 46.5 | 5.5 | 50.0 | **82.5** | 49.0 | 42.0 | 0.5 |
| | Flamingo | 45.5 | 46.0 | 56.0 | 39.5 | 35.5 | 49.0 | 47.0 | 9.0 | 53.0 | 80.0 | 43.0 | 29.5 | 1.0 |
| | Multimodal GPT | 51.0 | 45.5 | 9.5 | 7.0 | 0.5 | 45.5 | 45.0 | 0.0 | 65.0 | 81.5 | 32.0 | 5.0 | 0.0 |
| 20M | Ours | **100.0** | **100.0** | **100.0** | **100.0** | **61.0** | **100.0** | **100.0** | **16.5** | 75.5 | 75.0 | **96.0** | 37.5 | **47.5** |
| | Gato | 44.0 | 51.5 | 39.0 | 51.0 | 38.5 | 47.5 | 52.5 | 6.0 | 65.5 | **84.0** | 52.5 | 40.5 | 1.0 |
| | Flamingo | 48.5 | 49.0 | 55.5 | 48.0 | 42.5 | 46.5 | 52.0 | 6.0 | 66.0 | 82.0 | 47.5 | 37.0 | 0.5 |
| | Multimodal GPT | 50.5 | 49.5 | 53.0 | 44.5 | 43.5 | 47.0 | 46.0 | 8.0 | **83.5** | 80.0 | 46.5 | **41.0** | 2.5 |
| 200M | Ours | **100.0** | **100.0** | 99.5 | **100.0** | 54.5 | **100.0** | **100.0** | 17.5 | **77.0** | 93.0 | **98.5** | 75.0 | **45.0** |
| | Gato | 56.5 | 53.5 | 88.0 | 55.5 | 43.5 | 55.5 | 53.0 | 14.0 | 63.0 | 90.5 | 81.5 | 33.0 | 4.0 |
| | Flamingo | 51.0 | 52.5 | 61.5 | 49.5 | 38.5 | 47.5 | 55.5 | 5.5 | 70.5 | 82.0 | 42.0 | 39.0 | 3.0 |
| | Multimodal GPT | 52.0 | 52.0 | 49.5 | 54.5 | 45.5 | 52.5 | 51.0 | 11.0 | 76.5 | 84.0 | 43.0 | 38.0 | 0.5 |

Table 12: L3 level generalization results. Model indicates robot controller parameter count.

| Model | Method | Task 01 | Task 02 | Task 03 | Task 04 | Task 05 | Task 06 | Task 07 | Task 09 | Task 11 | Task 15 | Task 16 | Task 17 |
|---|---|---|---|---|---|---|---|---|---|---|---|---|---|
| 2M | Ours | **100.0** | **100.0** | **100** | **98.0** | **34.5** | **100.0** | **99.5** | **17.0** | **97.5** | **94.0** | **48.5** | **39.0** |
| | Gato | 45.5 | 48 | 28.0 | 23.0 | 3.0 | 45.5 | 45.0 | 2.5 | 40.5 | 29.5 | 37.0 | 1 |
| | Flamingo | 41.5 | 54.5 | 50.5 | 39.5 | 29 | 45.0 | 49.5 | 5.5 | 57.5 | 22.5 | 25.0 | 0.0 |
| | Multimodal GPT | 48.5 | 50.0 | 5.0 | 7.0 | 2.5 | 47 | 45.5 | 2.0 | 69.5 | 22.5 | 5.0 | 0.0 |
| 20M | Ours | **98.0** | **100.0** | **100** | **98.5** | **55.5** | **100.0** | **99.5** | **15.0** | 88.5 | **99.5** | 44.0 | **29.5** |
| | Gato | 46.5 | 55 | 44.5 | 57.0 | 31.5 | 47.5 | 51.5 | 2.5 | 72.5 | 30.5 | **44.0** | 0 |
| | Flamingo | 47.0 | 54.5 | 53.0 | 55.0 | 36 | 42.5 | 48.0 | 6.5 | 70.0 | 33.0 | 41.5 | 0.0 |
| | Multimodal GPT | 50.0 | 60.5 | 56.5 | 48.0 | 33.5 | 51 | 46.0 | 6.5 | **92.5** | 32.5 | 43.5 | 1.5 |
| 200M | Ours | **99.0** | **100.0** | **100** | **97.0** | 54.5 | **100.0** | **99.0** | 17.5 | 90.5 | **97.5** | **46.0** | **43.5** |
| | Gato | 51.0 | 58 | 84.5 | 56.5 | 35.5 | 53.5 | 49.0 | 15.0 | 65.0 | 52.0 | 33.0 | 0 |
| | Flamingo | 49.0 | 50.0 | 66.5 | 47.0 | 35 | 47.5 | 50.0 | 4.0 | 66.0 | 30.5 | 43.5 | 0.5 |
| | Multimodal GPT | 52.0 | 51.0 | 55.0 | 49.5 | 40.0 | 46 | 50.5 | 5.0 | 82.0 | 37.0 | 38.0 | 1.5 |

Table 13: L4 level generalization results. Model indicates robot controller parameter count.

| Model | Method | Task 08 | Task 10 | Task 13 | Task 14 |
|---|---|---|---|---|---|
| 2M | Ours | 6.5 | 0 | **0** | **96.5** |
| | Gato | 21.0 | **0.5** | **0** | 32 |
| | Flamingo | 22.0 | 0 | **0** | 27.5 |
| | Multimodal GPT | **22.5** | 0.0 | **0** | 22.0 |
| 20M | Ours | **100.0** | 0 | **0** | **95.5** |
| | Gato | 20.5 | 0.0 | **0** | 29 |
| | Flamingo | 21.0 | 0 | **0** | 27.5 |
| | Multimodal GPT | 20.5 | **0.5** | **0** | 36.0 |
| 200M | Ours | **100.0** | 0 | **0** | **94.5** |
| | Gato | 30.5 | **0.0** | **0** | 37 |
| | Flamingo | 24.5 | 0 | **0** | 24.0 |
| | Multimodal GPT | 20.0 | **0.0** | **0** | 28.5 |

**Imperfect Prompts**. We then study the policy robustness against imperfect prompts, including incomplete prompts (randomly masking out words with <UNK> token) and corrupted prompts (randomly swapping words, which could have changed the task meaning altogether). We ran our largest VIMA model with 200M parameters, results are shown in Table 16.

Table 14: Performances of our method with different sized pre-trained T5 prompt encoder. We fix the parameter count of the decision-making part to be 200M.

|    | `t5-small` (30M) | `t5-base` (111M) | `t5-large` (368M) |
|----|------|------|------|
| L1 | 78.8 | 81.5 | 80.8 |
| L2 | 79.0 | 81.5 | 81.0 |
| L3 | 80.3 | 78.7 | 81.0 |
| L4 | 49.1 | 48.6 | 49.3 |

Table 15: Evaluation results on tasks with increased amounts of distractors. We fix the parameter count of the decision-making part to be 200M.

|  | L1 | L2 | L3 | L4 |
|---|---|---|---|---|
| Original | 81.5 | 81.5 | 78.7 | 48.6 |
| More Distractors | 78.5 | 78.6 | 72.9 | 47.8 |
| Relevant Performance Decrease (%) | 3.6 | 3.5 | 7.3 | 1.6 |

Our well-trained model exhibits minimal performance decrease when evaluated on masked prompts and minor decrease on corrupted prompts. We attribute this robustness to the high-quality, pre-trained T5 language backbones.

Table 16: Evaluation results with incomplete and corrupted prompts. We fix the parameter count of the decision-making part to be 200M.

|  | L1 | L2 | L3 | L4 |
|---|---|---|---|---|
| Original | 81.5 | 81.5 | 78.7 | 48.6 |
| Incomplete Prompts | 80.8 | 81.1 | 77.0 | 48.0 |
| Corrupted Prompts | 78.2 | 78.1 | 73.8 | 45.3 |
| Relevant Performance Decrease w/ Incomplete Prompts (%) | 0.8 | 0.4 | 2.1 | 1.2 |
| Relevant Performance Decrease w/ Corrupted Prompts (%) | 4.2 | 4.3 | 6.6 | 7.2 |

## F EXTENDED RELATED WORK

In this section, we provide an extended review of related work as complementary to Section 2.

**Multi-task Learning by Sequence Modeling.** In NLP domain, the Natural Language Decathlon (McCann et al., 2018) adopts a consistent question-answering format for a suite of 10 NLP tasks. In computer vision, Mask R-CNN (He et al., 2017), UberNet (Kokkinos, 2016), and 12-in-1 (Lu et al., 2020) leverage a single backbone model with multiple independent heads for different tasks. UVim (Kolesnikov et al., 2022) is another unified approach for vision that uses a language model to generate the guiding code for a second model to predict raw vision outputs. In multimodal learning, numerous works (Lu et al., 2022; Wang et al., 2022a; Zellers et al., 2021; 2022; Buch et al., 2022; Fu et al., 2021; Yang et al., 2022) investigate the unification of image, video, audio, and/or language modalities to deliver multi-purpose foundation models, though most of which are not equipped with decision-making facilities. Perceivers (Jaegle et al., 2021b;a) propose an efficient architecture to handle general-purpose inputs and outputs. BEiT-3 (Wang et al., 2022c) performs masked data modeling on images, texts and image-text pairs to pre-train a backbone for various downstream tasks. MetaMorph (Gupta et al., 2022a) learns a universal controller over a modular robot design space.

**Foundation Models for Embodied Agents.** Embodied agent research (Duan et al., 2022; Batra et al., 2020; Ravichandar et al., 2020; Collins et al., 2021) is adopting the large-scale pre-training paradigm, powered by a collection of learning environments (Abramson et al., 2020; Shridhar et al., 2020; Savva et al., 2019; Puig et al., 2018; Team et al., 2021; Toyama et al., 2021; Shi et al.,

2017). From the aspect of **pre-training for better representations**, LaTTe (Bucker et al., 2022) and Embodied-CLIP (Khandelwal et al., 2021) leverage the frozen visual and textual representations of CLIP (Radford et al., 2021) for robotic manipulation. From the perspective of leveraging **transformer as agent architecture**, methods such as Dasari & Gupta (2020) and MOSAIC (Zhao et al., 2022) achieve superior performance in one-shot video imitation tasks. They both use the self-attention mechanism with auxiliary losses such as inverse dynamics loss (Dasari & Gupta, 2020) and contrastive loss (Zhao et al., 2022) to learn robot controllers. Our work differs from them mainly in three aspects: a) our method employs a transformer backbone to autoregressively predict actions; b) we utilize pre-trained language models (Raffel et al., 2020) and best practices from Tsimpoukelli et al. (2021) to learn policies conditioned on prompts with interleaved texts, images, and even videos; and c) while these works mainly focus on solving the single task of one-shot video imitation with highly customized objectives, conceptually simple but effective, our model is learned in a multi-task way with only the behavior cloning objective to solve a strict superset of tasks.

**Robot Manipulation and Benchmarks.** There are many prior works that are not mentioned in the main paper that study different robotic manipulation tasks, such as constraint satisfaction (Bharadhwaj et al., 2021), one-shot imitation (Paine et al., 2018; Huang et al., 2019; Aceituno et al., 2021; Zhao et al., 2022), and rearrangement (Liu et al., 2021; Ehsani et al., 2021; Gan et al., 2021; Stengel-Eskin et al., 2022). Multiple simulation benchmarks are introduced to study the above tasks: 1) **Indoor simulation environments**: Habitat (Savva et al., 2019; Szot et al., 2021) is equipped with a high-performance 3D simulator for fast rendering and proposes a suite of common tasks for assistive robots. AI2-THOR (Ehsani et al., 2021; Deitke et al., 2022) is a framework that supports visual object manipulation and procedural generation of environments. 2) **Tabletop environments**: Meta-World (Yu et al., 2019), RLBench (James et al., 2019), and SURREAL (Fan et al., 2018; 2019) are widely used simulator benchmarks studying robotics manipulation with tabletop settings. CausalWorld (Ahmed et al., 2021) is a benchmark for causal structure and transfer learning in manipulation, requiring long-horizon planning and precise low-level motor control. MOSAIC (Zhao et al., 2022) features a challenging benchmark built on top of Zhu et al. (2020) to evaluate one-shot imitation learning. It proposes a three-step test setting to evaluate the representational and generalization capability. Compared to it, ours supports a wide spectrum of manipulation tasks, including one-shot imitation learning. All these aforementioned simulators and benchmarks do not natively support task specification and prompting with multiple modalities.

