# OpenReview forum: "VIMA: General Robot Manipulation with Multimodal Prompts"
_ICLR.cc/2023/Conference — Submitted to ICLR 2023_

### Official Review · Reviewer_L5UP · 2022-10-24

**Confidence:** 4
**Correctness:** 2
**Technical Novelty And Significance:** 2
**Empirical Novelty And Significance:** 1
**Recommendation:** 3

**Clarity, Quality, Novelty And Reproducibility:**

**Clarity**

Much of the paper is clearly written, though the language can be improved, and a few parts are confusing.

For instance, in the statement of the second contribution:  “capable of multi-tasking”. This is abuse of the verb multi-tasking, which actually refers to switching rapidly from one task to another, as humans are able to do. Multi-tasking does not refer to an agent’s ability to learn to perform multiple tasks.

More detailed explanation is needed on the following topic:  “passing the resulted token sequence to a pre-trained T5”

**Quality**

VIMA-Bench appears to be a potentially valuable set of tasks. But the paper’s experimental comparisons to baselines are poor, for the reasons cited above.

**Novelty**

VIMA-Bench is somewhat novel, but the VIMA architecture is not, as explained above.

**Reproducibility**

Code is provided.


**Strength And Weaknesses:**

**Strengths**

VIMA-Bench is a flexible and useful environment for evaluation of multi-modal prompting, scalability, and generalization in a robotic setting. Interleaving language and image tokens in a single input sequence has shown recent promise, as with the Gato agent.

**Weaknesses**

VIMA’s performance is reliant on three critical design features:

- Visual **object tokens** from an off-the-shelf object detector.
- **Cross-attention** to the prompt sequence.
- A pre-defined set of **high-level actions**.

The **object tokens** are a crutch, giving VIMA effective access to the underlying state space, while baseline models are not given similar access. This disparity is more than enough to explain VIMA’s improved results over the baselines, and makes the comparisons unfair. The paper says “We hypothesize that the data efficiency can be attributed to VIMA’s object-centric representation, which is less prone to overfitting than learning directly from pixels in the low-data regime.” But it’s common knowledge that state representations allow greater sample efficiency than raw image observations. So there is nothing novel or surprising about taking advantage of object-level representations when available.

**Cross-attention** is one standard method of inputting a prompt sequence. Another standard method is to prepend the prompt to the single sequence, and rely on self-attention. Cross-attention is most appropriate when the prompt sequence (as in machine translation) contains a wealth of potentially variable information that strongly determines the correct output sequence. Recent models like Imagen have leveraged cross-attention in this way. Self-attention (over a single sequence) is most appropriate when the prompt sequence is less informative, and serves simply to identify the current task from among a trained set of tasks. Recent models like Gato have leveraged self-attention in this way. By construction, prompts in VIMA-Benchmark are highly variable, and strongly determine the correct outputs, and therefore call for cross-attention. Gato and DT were originally designed for the other type of prompt, and cannot be expected to perform as well on VIMA-Benchmark. But cross-attention could easily be added to Gato or DT. In fact, VIMA’s transformer architecture is equivalent to DT with cross-attention. There is nothing novel or surprising about taking advantage of cross-attention when it is called for.

VIMA’s **high-level actions** are severely limited to a pre-defined set including “pick-and-place” and “wipe”. The baseline agents are given access to the same high-level actions, so fair comparisions are at least possible. But the results are not indicative of how the agents would perform on more common low-level actions, like robotic joint torques. This makes the term “generalist robot agent” in the paper’s title especially inappropriate.

VIMA’s performance depends on these three components. The paper itself notes that good results “can only be achieved with both cross-attention and object token sequence representation”. So given the strong caveats explained above, the experimental results presented in the paper provide no reliable evidence for how VIMA compares to the baseline models.


**Summary Of The Paper:**

This paper presents an agent architecture called VIMA (for **Vi**suo**M**otor **A**ttention model), which is a standard transformer encoder-decoder pair. Multimodal inputs are encoded as a single sequence of interleaved text tokens and visual object tokens. Transformer outputs are mapped to actions, as in the Decision Transformer (DT). VIMA is evaluated on a new multi-modal benchmark, VIMA-Bench, consisting of 17 meta-tasks, and thousands of tabletop task instantiations arranged in a 4-level protocol of increasing difficulty. Baseline models (including Gato and DT) are also evaluated on VIMA-Bench. The results show that VIMA outperforms the baselines by consistently large margins.

**Summary Of The Review:**

VIMA-Bench is potentially useful. But because of the flaws in VIMA's comparisons to baseline agents, little can be concluded about VIMA's capabilities.

***** POST-DISCUSSION UPDATE *****

In their last response to me, the authors said:

**"Among these variants, we find that the recipe of object token + cross-attention is the best for multimodal-prompted robot agents through extensive experiments."**


This is a nice summary of the work’s key experimental finding. My concern is that both parts of this recipe are unremarkable:
- The object tokens come from a model (Mask R-CNN) trained on this task’s ground-truth state information, making it unlikely to scale well beyond VIMA-Bench. This limitation makes the object tokens a crutch, and readers already know that such crutches boost performance.
- A long line of prior work, starting with the original transformer paper of 2017, has shown that cross-attention is a powerful method for processing complex input spaces. Since VIMA-Bench’s input space is complex by construction, readers shouldn’t be surprised to see that cross-attention performs well here.

I still see value in VIMA-Bench itself, with its diversity of tasks. In fact, I would argue for acceptance of a *VIMA-Bench* paper submitted to a datasets and benchmarks track, as long as the models evaluated (including VIMA itself) were presented simply as baselines that the community could compare against in further work. In such a light, VIMA’s reliance on an unscalable vision processing pipeline would not pose a problem, and its usage of cross-attention would simply provide experimental verification of performance one might expect.

But instead of focusing on VIMA-Bench, this paper focuses on VIMA as a model for general robot manipulation, starting with the title itself:  **”VIMA: General Robot Manipulation…”** This is why I have to leave my scored unchanged, despite all the impressive work that has gone into the paper.

---

> ### Author Response · Authors · 2022-11-15
> **Response (2/2)**
>
> > VIMA’s high-level actions are severely limited to a pre-defined set including “pick-and-place” and “wipe”. [...] This makes the term “generalist robot agent” in the paper’s title especially inappropriate.
>
> Thanks for the suggestion. We have added an extra Limitations section in Sec. 7 and toned down the claim accordingly.
>
> However, we would like to highlight two points:
>
> 1. We inherit the same high-level action space from well-established prior works, such as Transporter [Zeng et al., 2020]. While pick-and-place and wipe seem simple, they do cover a wide range of tabletop manipulation tasks and are crucial to industrial use cases like warehouse robots [Yoon et al., 2003; Berscheid et al., 2020; Devin et al., 2020;  Song et al., 2020].
>
> 2. While VIMA currently uses these two actions, the algorithm design is general-purpose and does not make assumptions about the particular action choices. For example, VIMA would require only minimal modifications to support more low-level action spaces like joint-torque control.
>
> * Zeng et al., Transporter Networks: Rearranging the Visual World for Robotic Manipulation, CoRL 2020.
> * Yoon et al., Real-Time Tracking and Pose Estimation for Industrial Objects Using Geometric Features, ICRA 2003.
> * Berscheid et al., Self-supervised Learning for Precise Pick-and-place without Object Model, IEEE RA-Letter 2020.
> * Devin et al., Self-Supervised Goal-Conditioned Pick and Place, RSS 2020.
> * Song et al., Grasping in the Wild: Learning 6DoF Closed-Loop Grasping from Low-Cost Demonstrations, IEEE RA-Letter 2020.
>
> > More detailed explanation is needed on the following topic: “passing the resulted token sequence to a pre-trained T5”.
>
> The revised description is provided at L200.
>
> Prompt tokenization produces a sequence of interleaved textual and visual tokens. We then follow the practice in [Tsimpoukelli et al., 2021] and encode the prompt via a pre-trained T5 encoder [Raffel et al., 2020]. Since T5 has been pre-trained on large text corpora, VIMA inherits the semantic understanding capability and robustness properties. To accommodate tokens from new modalities, we insert MLPs between the non-textual tokens and T5. To prevent catastrophic forgetting, VIMA finetunes the last two layers of the language encoder with layer-wise learning rate decay [He et al., 2022] but freezes all other layers.
>
>
> * Tsimpoukelli et al., Multimodal Few-Shot Learning with Frozen Language Models, NeurIPS 2021.
> * Raffel et al., Exploring the Limits of Transfer Learning with a Unified Text-to-Text Transformer, JMLR 2020.
> *  He et al., Masked Autoencoders Are Scalable Vision Learners, CVPR 2022.

---

> > ### Comment · Reviewer_L5UP · 2022-11-18
> > **Thank you**
> >
> > I commend the authors for providing such extensive details in their responses and in their revised paper. After reading all of the other reviews and responses, I continue to be impressed by the amount of work that this paper represents!
> >
> > The new “limitations” paragraph in the conclusion provides a clear summary of reservations that I still have about the paper:
> >
> > “Limited action primitives (only pick-and-place and wipe for now)”, which could be addressed by “temporally-extended robot controllers”.
> >
> > Given such limited action primitives, performance on VIMA-Bench tells us very little about how a model would scale to a more standard robot action space, such as that used by Gato. The author response claims that “VIMA would require only minimal modifications to support more low-level action spaces like joint-torque control”, but the ease of implementation says nothing about how well that implementation would perform.
> >
> > “Reliance on **domain-finetuned Mask R-CNN** to provide object tokens”, which could be addressed by a “stronger vision pipeline”.
> >
> > While the Mask R-CNN’s object representation is not as reliable as ground-truth state information, it is *fine-tuned* on ground-truth state information (L161), and is not practical to extend (as noted by Reviewer DxBB) to more challenging environments. So I still see VIMA’s object tokens as a crutch that prevents us from drawing many conclusions about how VIMA would perform with a more extensible vision encoder, such as those used by the baseline agents Gato and Multimodal GPT.
> >
> > The paper’s new conclusion goes on to say that if these weaknesses were addressed, “VIMA could potentially scale to more challenging problems.” That is the key question, but this current work doesn’t help us answer it.
> >
> > As noted in my review, the prompts in VIMA-Bench, by their highly informative structure, call for cross-attention instead of the direct modeling used by the baseline agents. This puts those baseline agents (Gato and Multimodal GPT) at a distinct disadvantage on VIMA-Bench. And while Flamingo does use cross-attention, it was originally designed as a text generation model rather than as an agent, so the Flamingo Agent constructed in this work is suspect as a baseline model, especially given that this paper says nothing about how (or whether) its hyperparameters were tuned for this work.
> >
> > To summarize, VIMA’s two distinguishing features (cross-attention, and object representations from the Mask R-CNN fine-tuned with ground-truth information) are particularly well-suited for VIMA-Bench. So it is unremarkable that VIMA outperforms baseline agents that don’t share those advantages.

---

> > > ### Author Response · Authors · 2022-11-21
> > > **Response (2/2)**
> > >
> > > > [...] especially given that this paper says nothing about how (or whether) its hyperparameters were tuned for this work.
> > >
> > > We follow best practices to train Transformer models for all methods, including baselines. We also comprehensively tuned hyperparameters for the modified Flamingo agent to achieve the best performance. Model hyperparameters are provided in the supplementary (Tables 3, 4, 5, 6, and 8). Training hyperparameters are provided in Table 7.

---

> > > > ### Comment · Reviewer_L5UP · 2022-11-29
> > > > **Thank you again!**
> > > >
> > > > I appreciate the consistent thoroughness and clarity of all the author responses, greatly facilitating our discussion. In particular, many points made in their latest response highlight the weaknesses that I see with the paper.
> > > >
> > > > **"We note that sim2real is a widely adopted technique in both computer vision and robotics. Training with ground-truth information in simulation has been proven to improve performance and robustness in the real world."**
> > > >
> > > > While *instances* of such improvement have been reported in prior work, the phrase “proven to improve” seems too strong. I agree that achieving good sim-to-real transfer through training on ground-truth information in simulation on VIMA-Bench would be a valuable contribution, but this paper’s experiments are all in simulation.
> > > >
> > > > **"These prior works provide evidence that VIMA’s vision pipeline can scale to more challenging problems"**
> > > >
> > > > I agree that demonstrating a more scalable vision pipeline for VIMA would be a contribution. But without such experiments, the six prior works cited only provide a plausibility argument rather than actual evidence.
> > > >
> > > > **"a lifted observation space like object token can be a key factor to stronger generalization in more challenging problems"**
> > > >
> > > > True enough, it *can be*, but “stronger generalization in more challenging problems” is missing from this work.
> > > >
> > > > **"none of the baselines actually works out of the box under our formulation… They should thus be considered as “VIMA-Gato” and “VIMA-Flamingo” variants"**
> > > >
> > > > This is a good point. In my view, the paper would be stronger if it presented the baselines in exactly this toned-down light.
> > > >
> > > > **"Among these variants, we find that the recipe of object token + cross-attention is the best for multimodal-prompted robot agents through extensive experiments."**
> > > >
> > > >
> > > > This is a nice summary of the work’s key experimental finding. My concern is that both parts of this recipe are unremarkable:
> > > > - The object tokens come from a model (Mask R-CNN) trained on this task’s ground-truth state information, making it unlikely to scale well beyond VIMA-Bench. This limitation makes the object tokens a crutch, and readers already know that such crutches boost performance.
> > > > - A long line of prior work, starting with the original transformer paper of 2017, has shown that cross-attention is a powerful method for processing complex input spaces. Since VIMA-Bench’s input space is complex by construction, readers shouldn’t be surprised to see that cross-attention performs well here.
> > > >
> > > > I still see value in VIMA-Bench itself, with its diversity of tasks. In fact, I would argue for acceptance of a *VIMA-Bench* paper submitted to a datasets and benchmarks track, as long as the models evaluated (including VIMA itself) were presented simply as baselines that the community could compare against in further work. In such a light, VIMA’s reliance on an unscalable vision processing pipeline would not pose a problem, and its usage of cross-attention would simply provide experimental verification of performance one might expect.
> > > >
> > > > But instead of focusing on VIMA-Bench, this paper focuses on VIMA as a model for general robot manipulation, starting with the title itself:  **”VIMA: General Robot Manipulation…”** This is why I have to leave my scored unchanged, despite all the impressive work that has gone into the paper.

---

> > > > > ### Author Response · Authors · 2022-12-11
> > > > > **Response**
> > > > >
> > > > > Thank you again for the constructive feedback! We are glad that you find our response clear and thorough, and our multimodal prompting framework and VIMA-bench impressive.
> > > > >
> > > > > We acknowledge that you are concerned about our presentation with over-claiming issues. Most of the suggestions on the writing aspect came after the rebuttal phase, so we were unable to update the draft. But these writing issues are easy to address on our end. Specifically, we plan to rephrase the title and tone down the novelty claims on the VIMA model components (cross-attention and object tokens) during camera-ready revisions. We sincerely hope that these presentation issues will not detract from our paper’s merits.

---

> > > ### Author Response · Authors · 2022-11-21
> > > **Response (1/2)**
> > >
> > > Thank you again for the thoughtful comments! We really appreciate that you are impressed with the amount of work this paper presents. We sincerely hope you could consider raising your score, as we believe the current rating does not reflect the merits of our paper, given our contributions of a novel multimodal prompting framework, open-sourced large-scale benchmark, and the proposed VIMA method, as appreciated by the other reviewers. We will address your remaining concerns below:
> > >
> > > > While the Mask R-CNN’s object representation is not as reliable as ground-truth state information, it is fine-tuned on ground-truth state information (L161), and is not practical to extend
> > >
> > > We note that sim2real is a widely adopted technique in both computer vision and robotics. Training with ground-truth information in simulation has been proven to improve performance and robustness in the real world.
> > >
> > > For example, a robot agent with a perception system trained on generated semantic labels in a simulated household environment can be deployed to the real world without adaptation [Xia et al., 2018]. In computer vision, synthetic data also improves the generalization in tasks such as semantic segmentation for urban scenes [Ros et al., 2016], multi-object tracking [Gaidon et al., 2016], object detection [Tremblay et al., 2018a], object pose estimation for robotic grasping [Tremblay et al., 2018b], crowd counting [Wang et al., 2019], etc.
> > >
> > > These prior works provide evidence that VIMA’s vision pipeline can scale to more challenging problems even if we use ground-truth information during training. VIMA does not rely on any privileged information at inference time.
> > >
> > > * Xia et al., Gibson Env: Real-World Perception for Embodied Agents, CVPR 2018.
> > > * Ros et al., The SYNTHIA Dataset: A Large Collection of Synthetic Images for Semantic Segmentation of Urban Scenes, CVPR 2016.
> > > * Gaidon et al., Virtual Worlds as Proxy for Multi-Object Tracking Analysis, CVPR 2016.
> > > * Tremblay et al., Training Deep Networks with Synthetic Data: Bridging the Reality Gap by Domain Randomization, CVPR Workshop 2018a.
> > > * Tremblay et al., Deep Object Pose Estimation for Semantic Robotic Grasping of Household Objects, CoRL 2018b.
> > > * Wang et al., Learning from Synthetic Data for Crowd Counting in the Wild, CVPR 2019.
> > >
> > > > I still see VIMA’s object tokens as a crutch that prevents us from drawing many conclusions about how VIMA would perform with a more extensible vision encoder
> > >
> > > We would like to argue that a lifted observation space like object token can be a key factor to stronger generalization in more challenging problems. In prior literature (cited in our paper), there is extensive evidence that mid-level visual features are better than raw pixels for agent learning. For example, [Sax et al., 2018] demonstrate that using mid-level features like keypoints and object recognition can greatly improve visuomotor policy generalization over end-to-end methods. [Zhou et al., 2020] shows strong policy transfer across different simulators by conditioning the policy on semantic segmentation outputs.
> > >
> > > * Sax et al., Mid-Level Visual Representations Improve Generalization and Sample Efficiency for Learning Visuomotor Policies, 2018.
> > > * Zhou et al., Domain Adaptation Through Task Distillation, ECCV 2020.
> > >
> > > > This puts those baseline agents (Gato and Multimodal GPT) at a distinct disadvantage on VIMA-Bench
> > >
> > > We hope to reiterate that our main novelty lies in the formulation of multimodal prompts that unify a variety of robot manipulation tasks. In this regard, none of the baselines actually works out of the box under our formulation: Gato conditions only on partial expert trajectories and does not support flexible prompts; the original Flamingo model is not designed for embodied agents in Alayrac et al., 2022. They should thus be considered as “VIMA-Gato” and “VIMA-Flamingo” variants, rather than a literal comparison. Our adaptation of prior methods under the multimodal prompting framework is a non-trivial contribution in itself.
> > >
> > > We did not intentionally disadvantage the baselines, nor is that the point of the paper. In our ablation studies, we have included variants that are conceptually equivalent to “Gato with object tokens” (L322), “Gato’s image patch with cross-attention” (L303), “Multimodal GPT’s image encoder with cross-attention” (L305), and so on. Among these variants, we find that the **recipe of object token + cross-attention** is the best for multimodal-prompted robot agents through extensive experiments. We recommend this recipe over raw pixels and purely GPT decoder architecture to future practitioners. This recommendation will be very useful to researchers who wish to build upon our benchmark and framework.
> > >
> > > * Alayrac et al., Flamingo: a Visual Language Model for Few-Shot Learning, NeurIPS 2022.
> > >
> > > **<CONTINUED>**

---

> ### Author Response · Authors · 2022-11-15
> **Response (1/2)**
>
> Dear reviewer,
>
> Thank you for your detailed review and thoughtful comments. We are pleased to know that you find our VIMA-Bench flexible and useful, and our multimodal prompting framework promising. We have added new materials, revised the paper, and addressed your concerns below.
>
> > The object tokens are a crutch, giving VIMA effective access to the underlying state space, while baseline models are not given similar access. [...] So there is nothing novel or surprising about taking advantage of object-level representations when available.
>
> We agree that prior literature has demonstrated the superiority of object-centric representations [Florence et al., 2019; Sieb et al., 2019]. However, we would like to clarify:
>
> 1. The object representation does not grant our model access to the ground-truth state information because the inputs to VIMA are bounding box coordinates and cropped patches from raw RGB frames. Since these are computed by Mask-RCNN, the bounding boxes can be noisy, and the cropped image may have irrelevant pixels.
>
> 2. To the best of our knowledge, we are the first to study the incorporation of object tokens into a decision-making framework based on sequence modeling. The way we incorporate object-centric representations (bounding box coordinates and RGB crop) is also different from prior literature [Florence et al., 2019; Sieb et al., 2019].
>
> 3. In Sec. 6.3 (L282), we **did perform** the fair comparison that you mention. Given the **same object token input**, we ablate VIMA’s cross-attention against the decoder-only variant, which is conceptually the same as “Gato with object tokens”. VIMA is extremely parameter efficient and consistently outperforms this Gato variant in the hardest evaluation setting. We also experimented with an alternative way to process the object tokens via Perceiver Resampler. We find that using raw object tokens directly can achieve more than 40% more success than object-perceiver. This is not a trivial discovery because the Perceiver Resampler adds more model capacity and seems to be highly effective in Flamingo [Alayrac et al., 2022]. We believe that our findings are valuable to the community. We have revised the text to clarify the same.
>
> * Florence et al., Self-Supervised Correspondence in Visuomotor Policy Learning, IEEE RA-Letter 2019.
> * Sieb et al., Graph-Structured Visual Imitation, CoRL 2019.
> * Alayrac et al., Flamingo: a Visual Language Model for Few-Shot Learning, NeurIPS 2022.
>
> > Cross-attention is one standard method of inputting a prompt sequence. [...] There is nothing novel or surprising about taking advantage of cross-attention when it is called for.
>
> We acknowledge that cross-attention is a standard method and has been proven to be effective in many tasks [Saharia et al., 2022; Jaegle et al., 2021]. However, we did not intend to claim that the cross-attention mechanism alone is a novel component.
>
> Our main novelty lies in the formulation of multimodal prompts that unify a wide spectrum of robot manipulation tasks, and we demonstrate that cross-attention is a simple, effective, and scalable way to do task conditioning. In contrast, prior works [Stepputtis et al., 2020; Dasari and Gupta, 2020; Batra et al., 2020] require different and problem-specific techniques that do not generalize to the diverse scenarios covered by a single VIMA model.
>
> Our work is aligned with the motivations of T5 [Raffel et al., 2020] and Gato [Reed et al., 2022]. Specifically, T5 unifies many NLP tasks into a single text-to-text framework, while prior works require fragmented, domain-specific methods. In terms of architecture, T5 is simply a vanilla encoder-decoder transformer. Similarly, Gato’s key novelty is not to introduce a new architecture but to demonstrate that a multi-domain, multi-embodiment agent can be effectively trained by tokenizing highly diverse data sources and scaling up model capacity.
>
> * Saharia et al., Photorealistic Text-to-Image Diffusion Models with Deep Language Understanding, 2022.
> * Jaegle et al., Perceiver: General Perception with Iterative Attention, ICML 2021.
> * Stepputtis et al., Language-Conditioned Imitation Learning for Robot Manipulation Tasks, NeurIPS 2020.
> * Dasari and Gupta, Transformers for One-Shot Visual Imitation, CoRL 2020.
> * Batra et al., Rearrangement: A Challenge for Embodied AI, 2020.
> * Raffel et al., Exploring the Limits of Transfer Learning with a Unified Text-to-Text Transformer, JMLR 2020.
> * Reed et al., A Generalist Agent, TMLR 2022.
>
> **<CONTINUED>**

---

### Official Review · Reviewer_nbxK · 2022-10-24

**Confidence:** 3
**Correctness:** 4
**Technical Novelty And Significance:** 3
**Empirical Novelty And Significance:** 3
**Recommendation:** 6

**Clarity, Quality, Novelty And Reproducibility:**

### Clarity and Quality
- The paper is very easy to read, clearly written.

### Novelty
- Architecture and training schemes are not novel but prompting scheme and introduced benchmarks seem novel enough

### Reproducibility
- The authors promised to release everything


**Strength And Weaknesses:**

### Strengths
- Clear writing with nicely looking illustrations
- Exhaustive experiments
- Introducing an interesting benchmark

### Weaknesses
- Dependency on a separate object detection module. While it's shown to be effective as using oracle information in considered setups, it's not clear whether this can work in real-world scenarios or non-tabletop tasks as the paper promises to be a generalist robot agent. To add on this, I don't see this as a big weakness but it could be an interesting future direction to consider.
- Interesting results and analysis on a newly introduced benchmark. But comparison with the existing approaches [1,2] on other benchmarks [3] could further strengthen the claims made in the paper.
- MIssing details on baselines. From only reading the paper, it's a bit difficult to understand what's the main difference of the approach to considered baselines without reading the baseline papers in a detail.
- on Decision Transformer -- Trajectory Transformer [Janner et al., 2021] significantly differs from Decision Transformer [Chen et al., 2021] but the paper writes this as Decision Transformer (DT) (Chen et al., 2021; Janner et al., 2021). This should be corrected. Moreover, the main component of DT is to condition on desired returns and make the agent select the desired actions -- without this, the method that conditions on multimodal prompt cannot be called DT. It's just a behavior cloning agent conditioned on multimodal prompt with the GPT architecture.
- on cross-attention -- it's worthwile to cite relevant video prediction literature [4] that also utilizes cross-attention to reduce the compute cost when conditioning on previous frames (it seems that the architecture is almost the same, or very similar).

[1] Dasari, Sudeep, and Abhinav Gupta. "Transformers for one-shot visual imitation." In Conference on Robot Learning, pp. 2071-2084. PMLR, 2021.

[2] Mandi, Zhao, Fangchen Liu, Kimin Lee, and Pieter Abbeel. "Towards more generalizable one-shot visual imitation learning." In 2022 International Conference on Robotics and Automation (ICRA), pp. 2434-2444. IEEE, 2022.

[3] Zhu, Yuke, Josiah Wong, Ajay Mandlekar, and Roberto Martín-Martín. "robosuite: A modular simulation framework and benchmark for robot learning." arXiv preprint arXiv:2009.12293 (2020).

[4] Yan, Wilson, Yunzhi Zhang, Pieter Abbeel, and Aravind Srinivas. "Videogpt: Video generation using vq-vae and transformers." arXiv preprint arXiv:2104.10157 (2021).

**Summary Of The Paper:**

This paper presents VIMA, a robot agent that can act conditioned on multimodal prompts consisting of natural language and image objects. The architecture consists of self-attention and cross-attention layers and train the model to minimize behavior cloning loss. The main novelty of the paper is to propose a scheme for multimodal prompting, and the paper shows that the proposed approach can work well for various tasks with a single model.

**Summary Of The Review:**

The paper is written well, provides a thorough investigation of the proposed architecture in a very wide setups. I don't see major concerns with the paper but I'm reluctant to further increase my score as all the results are based on a newly introduced benchmarks and baselines which are not exactly clear how they are implemented. I'm willing to change my score up or down after the discussion during the rebuttal phase.

---

> ### Author Response · Authors · 2022-11-15
> **Response**
>
> Dear reviewer,
>
> Thank you for your detailed review, thoughtful comments, and positive evaluation! We are delighted that you find our paper very easy to read, experiments exhaustive, and multimodal prompting novel. We have added new materials, revised the paper, and answered your questions below.
>
> > Dependency on a separate object detection module. While it's shown to be effective as using oracle information in considered setups, it's not clear whether this can work in real-world scenarios or non-tabletop tasks as the paper promises to be a generalist robot agent. To add on this, I don't see this as a big weakness but it could be an interesting future direction to consider.
>
> Thanks for pointing out! We have added an extra Limitations section in Sec. 7 accordingly. VIMA’s algorithm design is general enough to support other visual preprocessing methods. For example, one solution is to switch to object detectors that are more robust and open-vocabulary, such as ViLD [Gu et al., 2022]. This would enable VIMA to transfer to real-world scenarios with minimal modification.
>
> * Gu et al., Open-vocabulary Object Detection via Vision and Language Knowledge Distillation, ICLR 2022.
>
> > Interesting results and analysis on a newly introduced benchmark. But comparison with the existing approaches [Dasari and Gupta, 2020; Zhao et al., 2022] on other benchmarks [Zhu et al., 2020] could further strengthen the claims made in the paper.
>
>
> Thanks for bringing up these works. We have cited them and added discussions in Sec. 2. [Dasari and Gupta, 2020; Zhao et al., 2022] are both exceptional works in one-shot visual imitation, which is a particular task category that VIMA supports, in addition to many other categories like visual constraint and visual goal. VIMA-Bench differs from [Zhu et al., 2020] because it is the first to introduce and systematically evaluate multimodal prompting for a wide range of robot manipulation tasks.
>
> * Dasari and Gupta, Transformers for One-Shot Visual Imitation, CoRL 2020.
> * Zhao et al., Towards More Generalizable One-shot Visual Imitation Learning, ICRA 2022.
> * Zhu et al., robosuite: A Modular Simulation Framework and Benchmark for Robot Learning, 2020.
>
> > Missing details on baselines. From only reading the paper, it's a bit difficult to understand what's the main difference of the approach to considered baselines without reading the baseline papers in detail.
>
> Thanks for bringing this up. To clarify the differences between these methods, we have added a table (Appendix, Sec. C.1, L1169) to compare the visual tokenizer, prompt conditioning, and observation token counts for our method and baselines. We have also updated the code submission to include all baseline implementations.
>
> In our revised paper, we provide more details for the Gato baseline at L1230 and the Flamingo baseline at L1245.
>
> > On Decision Transformer -- Trajectory Transformer [Janner et al., 2021] significantly differs from Decision Transformer [Chen et al., 2021] but the paper writes this as Decision Transformer (DT) (Chen et al., 2021; Janner et al., 2021). This should be corrected.
>
> We agree and have fixed it in the paper.  We renamed this method to “multimodal GPT agent.” The revised description is provided in Sec. 6.1 (L247) and Appendix, Sec. C.3.3 (L1256).
>
> * Chen et al., Decision Transformer: Reinforcement Learning via Sequence Modeling, 2021.
> * Janner et al., Offline Reinforcement Learning as One Big Sequence Modeling Problem, NeurIPS 2021.
>
> > On cross-attention -- it's worthwhile to cite relevant video prediction literature [Yan et al., 2021] that also utilizes cross-attention to reduce the compute cost when conditioning on previous frames (it seems that the architecture is almost the same, or very similar).
>
> Thanks for the suggestion. We have revised Sec. 5 (L216) to include this.
>
> The cross-attention design enjoys three advantages: 1) strengthened connection to prompt; 2) intact and deep flow of the original prompt tokens; and 3) better computational efficiency, as demonstrated in VideoGPT [Yan et al., 2021] as well.
>
> * Yan et al., VideoGPT: Video Generation using VQ-VAE and Transformers, 2021.

---

> > ### Comment · Reviewer_nbxK · 2022-11-22
> > **Response**
> >
> > Thanks for your detailed response! I also read other reviews, and agree with Reviewer L5UP in that it's still not clear the proposed method would *really* scale up to, or be effective on, other environments as it's only evaluated on newly introduced VIMA benchmark. I still think the paper could be a lot strengthened when the method can be compared with other prior baselines on standard benchmarkswith low-level controllers and more complexity (as Reviewer xPBt and L5UP suggested). So I'll defer finalizing my score until the discussion period ends.

---

> > > ### Author Response · Authors · 2022-11-25
> > > **Response**
> > >
> > > Thank you again for the thoughtful comments!
> > >
> > > First, we would like to emphasize that there is no existing benchmark for multi-modal prompts where we could evaluate VIMA. We developed VIMA-Bench based on the RAVENS benchmark [Zeng et al., 2020]. Our VIMA-Bench supports a much wider variety of tasks like one-shot video demonstration and visual constraints, which is already a huge advance compared to previous benchmarks. Moreover, papers that introduce new benchmarks typically compare proposed methods on their benchmarks, such as Fan et al., 2022; Zhu et al., 2020;  Shridhar et al., 2020; Savva et al., 2019. Hence, we believe our evaluation of VIMA is in line with existing best practices and literature.
> > >
> > > Second, none of the baselines actually works out of the box for multi-modal task specification: Gato conditions only on partial expert trajectories and does not support flexible prompts; the original Flamingo model is not designed for embodied agents in Alayrac et al., 2022. They should thus be considered as “VIMA-Gato” and “VIMA-Flamingo” variants, rather than a literal comparison. Our adaptation of prior methods under the multimodal prompting framework is a non-trivial contribution in itself. Through extensive experiments, we showcase that the VIMA recipe, i.e., using object tokens and cross-attention, works the best for multi-modal task specification. We believe that these results would be of significant interest to the broader community of decision making and robot learning.
> > >
> > > * Zeng et al., Transporter Networks: Rearranging the Visual World for Robotic Manipulation, CoRL 2020.
> > > * Fan et al., MineDojo: Building Open-Ended Embodied Agents with Internet-Scale Knowledge, NeurIPS 2022.
> > > * Zhu et al., robosuite: A Modular Simulation Framework and Benchmark for Robot Learning, 2020.
> > > * Shridhar et al., ALFRED: A Benchmark for Interpreting Grounded Instructions for Everyday Tasks, CVPR 2020.
> > > * Savva et al., Habitat: A Platform for Embodied AI Research, ICCV 2019.
> > > * Alayrac et al., Flamingo: a Visual Language Model for Few-Shot Learning, NeurIPS 2022.

---

> > > > ### Comment · Reviewer_nbxK · 2022-12-01
> > > > **Response**
> > > >
> > > > Thank you for your response. I'd like to note two things before moving into the final discussion phase with other reviewers and AC.
> > > > - I appreciate the value of a new benchmark as it's interesting to evaluate the ability to take multi-modal prompts and provided with exhaustive experimental results with several baselines. And that's why I voted for a weak acceptance of the paper.
> > > > - My thought after reading other reviews is that the paper should be toned-down as it's evaluated on relatively simple (as Reviewers xPBt and L5UP suggested) environments for it to be a "generalist robot agent" as the paper states. Unfortunately this has not yet been evidenced with further experiments during the rebuttal phase, so it's difficult for me to champion the paper to be accepted.

---

> > > > > ### Author Response · Authors · 2022-12-11
> > > > > **Thank you!**
> > > > >
> > > > > Thank you again for the constructive feedback! We are glad that you find our multimodal robot prompting framework novel and VIMA-bench impressive.
> > > > >
> > > > > We acknowledge that you are concerned about our presentation with over-claiming issues. Most of the suggestions on the writing aspect came after the rebuttal phase, so we were unable to update the draft. But these writing issues are easy to address on our end. We will gladly tone down in our camera-ready revisions. We sincerely hope that this feedback will not detract from our paper’s merits.

---

> ### Author Response · Authors · 2022-11-18
> **Follow-up on the response**
>
> Dear reviewer,
>
> As the discussion stage 1 is ending soon, we wonder if our response answers your questions and addresses your concerns? If yes, would you kindly consider raising the score? Thanks again for your very constructive and insightful feedback!

---

### Official Review · Reviewer_xPBt · 2022-10-24

**Confidence:** 4
**Correctness:** 3
**Technical Novelty And Significance:** 2
**Empirical Novelty And Significance:** 3
**Recommendation:** 5

**Clarity, Quality, Novelty And Reproducibility:**

+ The paper is clearly written.
- The novelty of the paper is the multimodal prompt input. However, the paper spends very little space explaining this novelty.

**Strength And Weaknesses:**

+ The paper reports on an impressive amount of work on the development of the agent and the benchmark. The 20+ pages of appendices are an impressive quantity of work.
- The paper proper only spends about 2/3 of a page (section 5) describing the model. This is unsufficient for understanding what exactly the contribution is.
- As a robot manipulation problem, the tasks are very simple. These are pick and place tasks, performed on a flat surface, in simulation, with objects that are clearly distinguishable.
- The paper does not explain why it considers that the multi-model prompt is superior. Are the prompts supposed to be generated by humans? Humans don't insert visual snippets or texture samples in the speech.
- The tasks in the tasks suite are specified in ways that are much simpler than what is the state of the art in this field. For instance, "novel councept grounding" appears to be just attaching a label to a definition. "one shot video imitiation" appears to be the exact reproducing of an exact trajectory - a much weaker definition that what is normally meant by imitation learning.

**Summary Of The Paper:**

The paper describes a technique to create a robot control agent that can control a manipulator to execute a variety of tasks specified in the form of a multi-model prompt, combining text and visual tokens.

The paper also introduces a large scale benchmark for problems focusing on the properties of the system VIMA-Bench.


**Summary Of The Review:**

The paper describes an impressive project, with a large number of demonstrations. However, it does not clearly focuses on what the novelty of the project is.

----

I have read the responses of the authors. They do not change my rating of the paper.

---

> ### Author Response · Authors · 2022-11-15
> **Response (2/2)**
>
> > The tasks in the tasks suite are specified in ways that are much simpler than what is the state of the art in this field. For instance, "novel concept grounding" appears to be just attaching a label to a definition. "One shot video imitation" appears to be the exact reproduction of an exact trajectory - a much weaker definition than what is normally meant by imitation learning.
>
> Thanks for voicing the concern. We hope to point out that these tasks share similar definition and complexity as some prior works. For example, [Hill et al., 2020] adopts the same definition of “novel concept grounding”, which assigns a new word to a previously unseen object. Then the new concept is immediately used in the grounded instructions. Our one-shot video imitation specification also closely follows [Huang et al., 2019a; Huang et al., 2019b; Dasari and Gupta, 2020; Zhao et al., 2022], where the robot agent is supposed to reproduce the trajectory from a single video demonstration.
>
> We acknowledge that more sophisticated versions of these tasks may exist. They can be added to our VIMA-Bench in future works and do not diminish the benchmark’s contribution, which is to systematically evaluate a single agent that solves diverse manipulation tasks conditioned on multimodal prompts.
>
> * Hill et al., Grounded Language Learning Fast and Slow, 2020.
> * Huang et al., Neural Task Graphs: Generalizing to Unseen Tasks from a Single Video Demonstration, CVPR 2019a.
> * Huang et al., Continuous Relaxation of Symbolic Planner for One-Shot Imitation Learning, IROS 2019b.
> * Dasari and Gupta, Transformers for One-Shot Visual Imitation, CoRL 2020.
> * Zhao et al., Towards More Generalizable One-shot Visual Imitation Learning, ICRA 2022.
>
> > The novelty of the paper is the multimodal prompt input. However, the paper spends very little space explaining this novelty.
>
> Thanks for the advice. We have expanded the writing in Sec. 3, Sec. 4, and Appendix Sec. B and discussed more on our contrast with prior works.

---

> ### Author Response · Authors · 2022-11-15
> **Response (1/2)**
>
> Dear Reviewer,
>
> We are grateful for your thoughtful comments and feedback! We are delighted that you find our benchmarking suite impressive and our paper clearly written. Following your suggestions, we have revised the paper and will address your remaining questions below.
>
> > The paper proper only spends about 2/3 of a page (section 5) describing the model. This is insufficient for understanding what exactly the contribution is.
>
> Thanks for the suggestion! We have revised Sec. 5 to present more details. The main contribution of our model architecture section is to provide practitioners with an effective recipe: prompt conditioning with cross-attention and object-centric tokenizers are the best combination for sequential robot manipulation.
> Moreover, we have also updated the code submission to include all method implementations, including baseline methods.
>
> In addition to the VIMA model, our primary novelty is the formulation of multimodal prompting for robotics. This is presented in Sec. 3 and Sec. 4 with great details.
>
> > As a robot manipulation problem, the tasks are very simple. These are pick and place tasks, performed on a flat surface, in simulation, with objects that are clearly distinguishable.
>
> We acknowledge that the simulation is relatively clean, and the current tasks are in a tabletop environment. We added an extra Limitations section in Sec. 7 accordingly. However, we would like to highlight that our novelty lies primarily in the introduction of multimodal prompts that unify a variety of robot manipulation tasks. While VIMA is currently trained for simulated tabletop tasks, the algorithm design is general-purpose and does not make assumptions about the particular observation and action choices. In future work, VIMA can be scaled to more realistic and complex environments with minimal modifications, such as replacing the Mask R-CNN with more powerful object detectors [Gu et al., 2022] and expanding the action space to more complex primitives.
>
> * Gu et al., Open-vocabulary Object Detection via Vision and Language Knowledge Distillation, ICLR 2022.
>
> > The paper does not explain why it considers that the multi-modal prompt is superior. Are the prompts supposed to be generated by humans? Humans don't insert visual snippets or texture samples in the speech.
>
> We consider multimodal prompting superior because it enables a single model to learn different task specification interfaces, such as language instruction [Stepputtis et al., 2020], one-shot video demonstration [Duan et al., 2017; Finn et al., 2017], and goal image [Ding et al., 2019]. Each of these interfaces is extensively studied in the literature and has complementary pros and cons in different situations. VIMA is a single backend model that can be deployed to all these scenarios without re-training for each. Whether it is intuitive to use for humans, however, depends on the frontend UX design. It is an orthogonal question that would be interesting to explore in the HCI field.
>
> * Stepputtis et al., Language-Conditioned Imitation Learning for Robot Manipulation Tasks, NeurIPS 2020.
> * Duan et al., One-Shot Imitation Learning, NeurIPS 2017.
> * Finn et al., One-Shot Visual Imitation Learning via Meta-Learning, CoRL 2017.
> * Ding et al., Goal-conditioned Imitation Learning, NeurIPS 2019.
>
> **<CONTINUED>**

---

> ### Author Response · Authors · 2022-11-18
> **Follow-up on the response**
>
> Dear reviewer,
>
> As the discussion stage 1 is ending soon, we wonder if our response answers your questions and addresses your concerns? If yes, would you kindly consider raising the score? Thanks again for your very constructive and insightful feedback!

---

### Official Review · Reviewer_DxBB · 2022-10-27

**Confidence:** 5
**Correctness:** 4
**Technical Novelty And Significance:** 3
**Empirical Novelty And Significance:** 3
**Recommendation:** 8

**Clarity, Quality, Novelty And Reproducibility:**

Clarity:
The paper is extremely clear and well written. The sections are well organize. The methodology is well done and the ablation studies well selected. The related work is extensive and completely relevant for this work.

Quality:
The paper quality also is superb. The content of the theory, the figures, the explanation and text provide a very high quality paper. The completenees of the work with the available code, the extent of the work on the dataset are very thorough and extensive.

Reproducibility:
The code is available and seems well done and functional. I have not run it.

Novelty: This work even if aligned to GATO as well describe in the text, is novel as pointed out on the different point on the strengths section before.

**Strength And Weaknesses:**

Strengths:
1) Novel transformer architecture for robot manipulation.
2) Novel multimodal interface for robot human interaction.
3) Novel insight on improved performance on using object centric approaches for this tasks.
4) New dataset for multimodal and multitask robot manipulation.
5) Novel methodology to measure generalization in robot manipulation.
6) Novel human robot interaction modality.
Weaknesses:
1) Unclear how this method will generalize to temporal extended actions : i.e, working the dough, beating eggs, writing, folding clothes, etc.
2) Finteunning Mask-RCNN seems that might not be practical if we want to extend this method to work in-the-wild. What's the accuracy for a non-finetune Mask-RCNN?
3) What will happen if the user change his mind after saying the command?
4) How is uncertainty deal in this model? How could be resolved?
5) What would happen if you have a cluttered scene and the number of object grow and there some that irrelevant to task?
6) What would happened if there is something incomplete, or wrong in the text command?
7) It could be interesting if also the model can output text to resolved some of the aforementioned items.


**Summary Of The Paper:**

This research develops a novel multimodal , multitask architecture for robot manipulation. Also introduces a multimodal and multitask benchmark for robot manipulation. Finally introduce a carefully thought methodology in 4 steps to quantify the capability of the developed architecture.  The results section provide relevant ablation studies that bring insights on this line of research, providing ideas to further improve this research domain.

**Summary Of The Review:**

A large effort on developing a novel human robot interaction interface on robotics manipulation tasks that generalize. The work provide a relevant datase, new methodology to measure generalization, and novel models to solved this multitask and multimodal tasks.

---

> ### Author Response · Authors · 2022-11-15
> **Response (2/2)**
>
> > What would happen if there is something incomplete, or wrong in the text command?
>
> Great question! We have done an extra evaluation experiment with incomplete prompts (randomly masking out words with `<UNK>` token) and corrupted prompts (randomly swapping words, which could have changed the task meaning altogether). For our largest 200M model, we get the following success rates:
>
> |                                                         | L1   | L2   | L3   | L4   |
> |--------------------------------------------------------:|------|------|------|------|
> |                                                Original | 81.5 | 81.5 | 78.7 | 48.6 |
> |                                      Incomplete Prompts | 80.8 | 81.1 | 77.0 | 48.0 |
> |                                       Corrupted Prompts | 78.2 | 78.1 | 73.8 | 45.3 |
> | Relevant Performance Decrease w/ Incomplete Prompts (%) | 0.8  | 0.4  | 2.1  | 1.2  |
> |  Relevant Performance Decrease w/ Corrupted Prompts (%) | 4.2  | 4.3  | 6.6  | 7.2  |
>
> We observe minimal performance decrease on masked prompts and minor decrease on corrupted prompts. We attribute this robustness to the high-quality, pretrained T5 language backbones.
>
> > It could be interesting if the model can output text to resolve some of the aforementioned items.
>
> Absolutely, this is an interesting direction to explore! We may use models like BERT and FCM [Devlin et al., 2018; Liu et al., 2022] to infill the incomplete prompts and correct the corrupted ones.
>
> * Devlin et al, BERT: Pre-training of Deep Bidirectional Transformers for Language Understanding, 2018.
> * Liu et al., FCM: Forgetful Causal Masking Makes Causal Language Models Better Zero-Shot Learners, 2022.

---

> ### Author Response · Authors · 2022-11-15
> **Response (1/2)**
>
> Dear Reviewer,
>
> Thank you so much for your thoughtful review and positive evaluation! We are glad that you find our multimodal prompt interface, agent architecture, and benchmark novel. We have revised the paper following your suggestions, and will address your remaining questions below.
>
> > Unclear how this method will generalize to temporal extended actions : i.e, working the dough, beating eggs, writing, folding clothes, etc.
>
> This is an interesting suggestion for follow-up works! Since VIMA makes no assumptions about the specific action primitives, we can extend the current action spaces to support temporally extended actions like those in [Nasiriany et al., 2022].
>
> *  Nasiriany et al., Augmenting Reinforcement Learning with Behavior Primitives for Diverse  Manipulation Tasks, ICRA 2022.
>
> > Fine-tuning Mask-RCNN seems that might not be practical if we want to extend this method to work in-the-wild. What's the accuracy for a non-finetune Mask-RCNN?
>
> We acknowledge that VIMA might not work well without finetuning in the wild. We have added an extra Limitations section in Sec. 7 to explain this. One solution is to switch to more powerful object detectors that are robust in the wild and support open vocabulary, such as ViLD [Gu et al., 2022].
>
> Following your suggestion, we construct an offline test set of scene images to investigate the performance of a non-finetuned Mask-RCNN. We use IoU between predicted and ground-truth bounding boxes as the metric. We found that IoUs for finetuned and non-finetuned Mask-RCNN are 95.9% and 78.1%, respectively, suggesting the necessity of the finetuning stage.
>
> * Gu et al., Open-vocabulary Object Detection via Vision and Language Knowledge Distillation, ICLR 2022.
>
> > What will happen if the user changes his mind after saying the command?
>
> This is an interesting idea! The current VIMA model does not take extra user input during the rollout. One potential solution is to follow the recipe in [Huang et al., 2022] to support human feedback in a loop.
>
> * Huang et al., Inner Monologue: Embodied Reasoning through Planning with Language Models, 2022.
>
> > How is uncertainty dealt in this model? How could this be resolved?
>
> We discretize the action space and use softmax classification loss for training instead of regression. This recipe follows prior works in embodied agent learning [Metz et al., 2017; Shafiullah et al., 2022] to better model multimodal action distribution.
>
> In our benchmark, there are many tasks that have non-unique solutions. For example, the prompt template for a visual constraint satisfaction task is “Sweep any {object} into {container} without touching {constraint}”. Because the oracle randomly selects one object to move, our VIMA model trained by imitation will be able to learn a multimodal action distribution from the data.
>
> * Metz et al., Discrete Sequential Prediction of Continuous Actions for Deep RL, 2017.
> * Shafiullah et al., Behavior Transformers: Cloning K modes with one stone, NeurIPS 2022.
>
> > What would happen if you have a cluttered scene and the number of object grows and some are irrelevant to the task?
>
> We would like to clarify that (1) typical scenes are already quite cluttered in tasks like Novel Concept Grounding, Visual Constraint Satisfaction, and Visual Reasoning; (2) there is at least one distractor object for each task in VIMA-Bench.
>
> To answer your question more quantitatively, we perform extra evaluations of our trained models on more distractors for all tasks. We obtain the following success rates for our largest 200M model:
>
> |                                   | L1   | L2   | L3   | L4   |
> |----------------------------------:|------|------|------|------|
> |                          Original | 81.5 | 81.5 | 78.7 | 48.6 |
> |                  More Distractors | 78.5 | 78.6 | 72.9 | 47.8 |
> | Relevant Performance Decrease (%) | 3.6  | 3.5  | 7.3  | 1.6  |
>
> We observe that the performance of VIMA degrades minimally with more distractors than the training distribution. This indicates that our agent has learned a reasonably robust policy.
>
> **<CONTINUED>**

---

> ### Author Response · Authors · 2022-11-18
> **Follow-up on our response**
>
> Dear reviewer,
>
> As the discussion stage 1 is ending soon, we wonder if our response answers your questions and our extra experiments address your concerns? If yes, would you kindly consider raising the score? Thanks again for your very constructive and insightful feedback!

---

### Author Response · Authors · 2022-11-15
**Thank you to all reviewers and meta-reviewers!**

We sincerely thank all reviewers for their thoughtful and constructive feedback. We really appreciate that many reviewers found our multimodal prompting framework novel, our new VIMA-Bench simulation suite impressive, our model architecture effective, and our experiments and ablations extensive and thorough.

For this rebuttal, we have conducted many extra evaluations and analyses to address the feedback. We have updated the code submission to include **all** baseline implementations to facilitate better understanding and evaluation of our contributions. In our response to each reviewer below, we address their individual questions and comments. The paper and supplementary PDFs have been updated with revisions, highlighted in yellow. We welcome any follow-up discussions!

---

### Decision · Program_Chairs · 2023-01-20

**Decision:**

Reject

**Justification For Why Not Higher Score:**

The paper stays very controversial. 2 reviewers could not be convinced that the proposed method is good enough. Having a major refocus on the benchmark would be OK for most reviewers, but I don't think is advisable in this stage.

**Justification For Why Not Lower Score:**

N/A

**Metareview: Summary, Strengths And Weaknesses:**

Summary:
The paper proposes a new approach enabling multi-modal prompts (images + text) in a transformer-based agent. The authors also propose a novel benchmark for this setting.

Strength:
- First paper enabling this interesting setting
- Good experimental results
- Useful benchmark

Weaknesses:
- Neither fish nor meat paper
- Doubts about the method and its novelty

Additional comments after re-reading the paper:
- domain-finetuned Mask R_CNN: unclear which data/objects are used for that
- the single objects in the example prompts in the paper and appendix are always canonical views - bringing them somewhat closer to words, this doesn't seem to be explained

**Summary Of Ac-Reviewer Meeting:**

All 4 reviewers attended. We discussed extensively, which was helpful for clarifying positions, but didn't come any closer to an agreement - i.e. the reviewers were still ranging from strong reject, weak reject, weak accept, strong accept.

The positive camp recognizes that the ingredients are not novel, but like the extensive work and careful replies, overall finding VIMA reasonable. The negative camp is concerned about the required object model, which the other side doesn't consider a major problem if it is available.

The negative camp was arguing for completely refocusing the paper on the benchmark. I do not agree with the last message by the authors that this is "easy to address" and would rather require a re-review. Some other reviewers are also not entirely convinced by the benchmark (i.e., might not be strong enough for a stand-alone paper) and believe the strength of the paper is actually in the combination.

There was also some disagreement on whether the method is described in sufficient detail in the paper (missing description on what is done with transformers) and whether that can be compensated for by code availability.

The visual constraint modality and grounding are novel and very interesting to see. Everybody agreed that this is a very interesting problem which would lead to good discussions at a conference - but some reviewers maintain that readers will be shaking heads about the model.

What we agreed on:
Generalization is a rather specific definition in this paper but OK (but should be removed from the title)